# Let Language Constrain Geometry:
# Vision–Language Models as Semantic and Spatial Critics for 3D Generation

**Weimin Bai** [1]  **Yubo Li** [1]  **Weijian Luo** [2]  **Zeqiang Lai** [3]  **Yequan Wang** [4]  **Wenzheng Chen** [1 4]  **He Sun** [1]

## Abstract

Text-to-3D generation has advanced rapidly, yet state-of-the-art models, encompassing both optimization-based and feed-forward architectures, still face two fundamental limitations. First, they struggle with coarse semantic alignment, often failing to capture fine-grained prompt details. Second, they lack robust 3D spatial understanding, leading to geometric inconsistencies and catastrophic failures in part assembly and spatial relationships. To address these challenges, we propose VLM3D, a general framework that repurposes large vision-language models (VLMs) as powerful, differentiable semantic and spatial critics. Our core contribution is a dual-query critic signal derived from the VLM's "Yes/No" log-odds, which assesses both semantic fidelity and geometric coherence. We demonstrate the generality of this guidance signal across two distinct paradigms: (1) As a reward objective for optimization-based pipelines, VLM3D significantly outperforms existing methods on standard benchmarks. (2) As a test-time guidance module for feed-forward pipelines, it actively steers the iterative sampling process of SOTA native 3D models to correct severe spatial errors. VLM3D establishes a principled and generalizable path to inject the VLM's rich, language-grounded understanding of both semantics and space into diverse 3D generative pipelines. Code is available at https://ai4scientificimaging.org/VLM3D/.

## 1. Introduction

Creating 3D content from natural language descriptions has drawn significant attention (Hunyuan3D et al., 2025; Zhao et al., 2025; Zhang et al., 2024; Poole et al., 2023; Shi et al., 2023). Recent models, encompassing both optimization-based paradigms (Poole et al., 2023) and high-speed feed-forward architectures (Hunyuan3D et al., 2025; Zhao et al., 2025; Zhang et al., 2024), have achieved impressive results. However, state-of-the-art models across both paradigms still face two fundamental limitations. First, they struggle with coarse semantic alignment, often failing to capture fine-grained prompt details or complex interactions. For instance, given a detailed description of the "Embracing Peace" monument, even the powerful MVDream baseline omits one of the two figures entirely (see Figure 1 top). Second, they lack robust 3D spatial understanding. For instance, SOTA feed-forward models (Hunyuan3D et al., 2025; Zhao et al., 2025; Zhang et al., 2024) can fail in part assembly and spatial relationships, generating nonsensical, floating collections of parts instead of coherent objects (see Figure 1 bottom). These failures arise from inadequate priors and limited reasoning: optimization-based methods inherit weak semantic grounding and poor spatial awareness from 2D diffusion models, while feed-forward methods are constrained by the limited complexity of 3D training data (often single objects rather than multi-object scenes), leading to poor modeling of 3D structure and compositional semantics. Both ultimately lack a deep understanding of spatial relationships and language.

To address these challenges across paradigms, we propose VLM3D, a 3D generation framework that integrates large vision-language models (VLMs) (Bai et al., 2023; 2025b) as powerful, differentiable semantic and spatial critics. Instead of relying on the weak semantic grounding of CLIP-style encoders (Radford et al., 2021b; Zarei et al., 2024) or the 2D-centric priors of diffusion models (Rombach et al., 2022; Shi et al., 2023), we leverage the VLM's rich, language-grounded understanding of both fine-grained semantics and complex 3D spatial relationships (Cheng et al., 2024; Chen et al., 2024; Zhang et al., 2025).

Our core contribution is a dual-query VLM-based critic signal that simultaneously optimizes the 3D representation

[1]Peking University, Beijing [2]hi-lab, Xiaohongshu Inc. [3]MMLab, CUHK [4]Beijing Academy of Artificial Intelligence. Correspondence to: He Sun <hesun@pku.edu.cn>.

*Proceedings of the 43rd International Conference on Machine Learning*, Seoul, South Korea. PMLR 306, 2026. Copyright 2026 by the author(s).

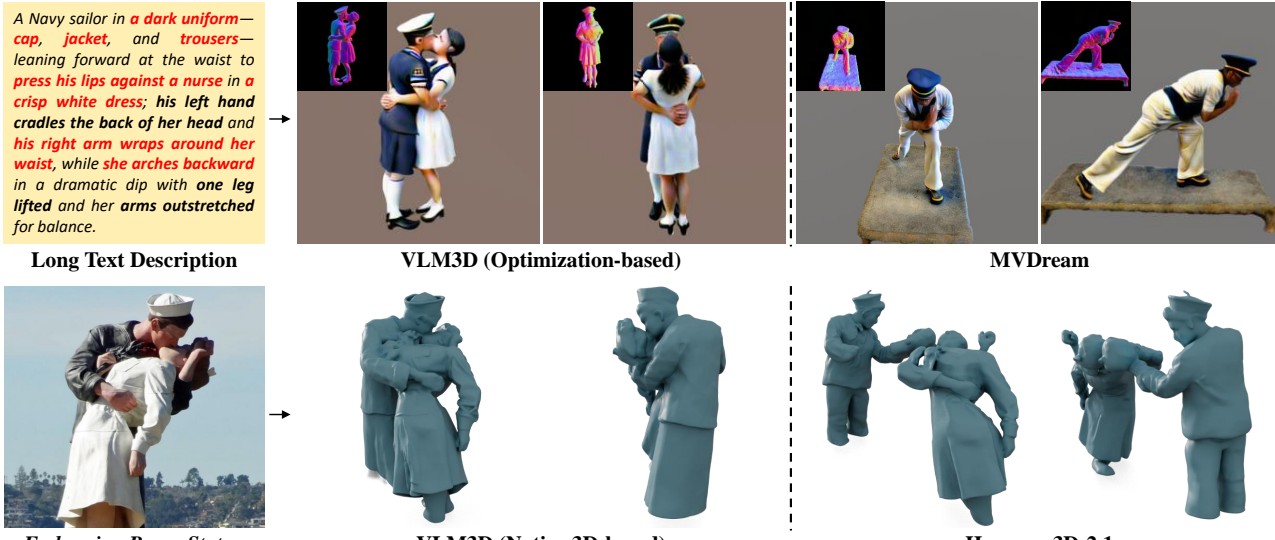

*A Navy sailor in a dark uniform— cap, jacket, and trousers— leaning forward at the waist to press his lips against a nurse in a crisp white dress; his left hand cradles the back of her head and his right arm wraps around her waist, while she arches backward in a dramatic dip with one leg lifted and her arms outstretched for balance.*

**Long Text Description**     **VLM3D (Optimization-based)**     **MVDream**

*Embracing Peace Statue*     **VLM3D (Native 3D-based)**     **Hunyuan3D-2.1**

*Figure 1.* **Reproducing the "Embracing Peace" Statue with VLM3D.** We challenge VLM3D's dual paradigms with San Diego's iconic monument. **Top (Optimization-based):** Given a long text description, baseline MVDream (Shi et al., 2023) suffers a catastrophic semantic failure, omitting the nurse entirely. Our VLM3D critic successfully recovers both figures and their signature pose. Key details are highlighted in red. **Bottom (Feed-forward model-based):** Comparing to the statue's reference image, baseline Hunyuan3D (Hunyuan3D et al., 2025) generates a distorted, spatially incoherent mesh. Our VLM3D substantially corrects these spatial failures and produces a more geometrically plausible 3D asset.

for both semantic fidelity and spatial/geometric coherence. Specifically, our framework tasks the VLM to act as a dual-objective reward model. It evaluates the multi-view renderings against a structured query comprising two distinct criteria: (1) Content Match: assessing correspondence to the semantic description, and (2) Geometric Quality: assessing whether the object is geometrically sound and consistent across views. The VLM is constrained to output only *Yes* or *No*; we then design our differentiable VLM reward based on the extracted log-odds. Backpropagating this signal provides a direct semantically-grounded and spatially-aware gradient for optimizing the 3D representation.

We demonstrate the generality of the proposed VLM-based critic by applying it across two distinct paradigms:

• Score Distillation Sampling: We integrate VLM3D into an optimization-based pipeline SDS (Poole et al., 2023; Shi et al., 2023) . On standard benchmarks like GPTEval3D (Wu et al., 2024a), VLM3D significantly outperforms existing methods in semantic accuracy and 3D plausibility. It successfully captures complex semantics (Fig. 1) and resolves geometric inconsistencies (Fig. 7).

• Feed-forward Models: We show that VLM3D can be deeply integrated into the iterative sampling loop of SOTA native 3D feed-forward models (Hunyuan3D et al., 2025; Zhao et al., 2025). As shown in Fig. 4, our test-time guidance approach effectively corrects severe geometric and part-assembly errors, producing coherent 3D assets where the original models failed.

To the best of our knowledge, VLM3D makes the first attempt to establish a principled and generalizable path to inject the VLM's rich, language-grounded understanding of both semantics and space into both optimization-based and feed-forward-based 3D generative pipelines.

## 2. Related Works

Our work builds upon the intersection of two major paradigms in text-to-3D generation and the rapid advancements in vision-language models.

### 2.1. Optimization-based Text-to-3D Generation

The first dominant paradigm leverages 2D diffusion priors (Ho et al., 2020; Rombach et al., 2022) to guide the per-scene optimization of a 3D representation. SDS (Poole et al., 2023) pioneered this by using the gradients from a pretrained 2D diffusion model to optimize a Neural Radiance Field (NeRF) (Mildenhall et al., 2020). This distillation process enables plausible 3D asset creation from text without 3D data (Poole et al., 2023).

Subsequent methods improved this foundation. Magic3D introduced a coarse-to-fine optimization strategy (Lin et al., 2023). MVDream trained a diffusion model on multi-view data to produce more 3D-consistent priors (Shi et al., 2023), while ProlificDreamer framed the process as variational inference to improve diversity (Wang et al., 2023). Other works have focused on integrating explicit reward models, such as DreamReward, which uses a 3D reward model

trained on human feedback (Ye et al., 2024).

Despite their success, these methods inherit the limitations of their 2D priors. Their guidance signals often lack fine-grained semantic understanding, relying on CLIP-style encoders (Radford et al., 2021a; Raffel et al., 2020) that struggle with complex prompts. Furthermore, being 2D-native, these priors lack explicit 3D spatial reasoning, leading to view-inconsistencies like the Janus problem (Wang et al., 2024c; Nath et al., 2025; Hong et al., 2023a; Seo et al., 2024). Our work addresses these semantic and spatial gaps with a VLM-based reward.

## 2.2. Feed-Forward Text-to-3D Generation

The second major paradigm is feed-forward models, developed to overcome the slow optimization of SDS. These methods aim to generate 3D assets in a single forward pass, enabling near real-time creation. Early approaches focused on generating explicit representations like point clouds or voxels. More recently, methods like Instant3D (Li et al., 2023b), LRM (Hong et al., 2023b) and GRM (Xu et al., 2024) first generate consistent multi-view images and then use a fast reconstruction model to produce NeRF or 3D Gaussian Splatting (Kerbl et al., 2023) assets.

Note, recent native 3D diffusion models (Hunyuan3D et al., 2025; Zhao et al., 2025; Zhang et al., 2024), which we also categorize as feed-forward, require an iterative sampling process to generate the final asset. These SOTA feed-forward models are trained on large-scale 3D datasets to directly output 3D representations from text or images. While extremely efficient, this speed can come at the cost of fidelity and coherence. These challenges are often attributable to the inherent limitations of current 3D training data, which remains relatively scarce and is largely constrained to single objects, lacking complex, multi-object scenes. Consequently, as demonstrated in Fig. 1, these models can exhibit notable limitations in spatial intelligence, such as generating disconnected parts, incomplete geometry, or incorrect spatial relationships. They also struggle with capturing the precise, fine-grained semantic attributes specified in a prompt. This highlights a critical need for a refinement mechanism. Our work positions VLM3D as a general test-time guidance module to address exactly these semantic and spatial failures.

## 2.3. Vision-Language Models

Our solution is powered by modern vision-language models. Early VLMs like CLIP (Radford et al., 2021b; Jia et al., 2021; Bao et al., 2021), while foundational, were trained with contrastive learning. This discriminative objective limits their generative flexibility (Yu et al., 2022; Li et al., 2024; Ramesh et al., 2021) and, crucially, their fine-grained spatial reasoning capabilities (Chen et al., 2024; Qiu et al., 2025;

Wang et al., 2024a; Patel et al., 2024).

Modern large VLMs extend this foundation by integrating a powerful autoregressive language model with a visual encoder in an end-to-end, generative framework. Methods like BLIP-2 (Li et al., 2023a) and MiniGPT-4 (Zhu et al., 2023) leverage lightweight query modules or adapters to bridge frozen vision and language backbones(Li et al., 2023a; Zhu et al., 2023), then fine-tune on multimodal instruction data to support open-ended tasks such as image captioning, visual question answering, dense grounding, and dialogue(Liu et al., 2023). More recently, models such as LLaVA (Liu et al., 2023; Sun et al., 2023) and Qwen-VL (Bai et al., 2025b) have demonstrated advanced spatial grounding capabilities—localizing objects, understanding complex relations, and reasoning over multi-object scenes—while retaining strong semantic alignment. Notably, these models incorporate advanced vision modules—for example, Qwen2.5-VL employs dynamic resolution processing to natively handle variable-size images and absolute time encoding for precise long-range video reasoning —further enhancing their spatiotemporal understanding(Bai et al., 2025b; Wang et al., 2024b). Because these VLMs provide language-grounded similarity measures and implicitly encode spatial relationships, they serve as ideal reward functions within text-to-3D generation pipelines(Poole et al., 2023; Hunyuan3D et al., 2025; Zhao et al., 2025; Zhang et al., 2024).

## 3. Preliminary

**2D Text-to-Image Diffusion Models**  Diffusion models (Ho et al., 2020; Sohl-Dickstein et al., 2015; Song et al., 2020) define a forward–time SDE that gradually injects noise into a data sample and a corresponding reverse–time SDE that removes noise to generate samples. Concretely, for an image $\mathbf{x}_0$ conditioned on text prompt $y$ (i.e., $\mathbf{x}_0 \sim p_{\text{data}}(\cdot \mid y)$ ), the forward SDE is

$$d\mathbf{x}_t = f(\mathbf{x}_t, t)\, dt + g(t)\, d\mathbf{w}_t, \qquad (1)$$

and the reverse–time SDE is

$$d\mathbf{x}_t = \left[ f(\mathbf{x}_t, t) - g(t)^2 \nabla_{\mathbf{x}_t} \log p_t(\mathbf{x}_t \mid y) \right] dt + g(t)\, d\bar{\mathbf{w}}_t, \qquad (2)$$

where $t \in [0, T]$, $f$ is the drift, $g$ the diffusion coefficient, $\mathbf{w}_t$ and $\bar{\mathbf{w}}_t$ are forward and reverse Wiener processes, and $p_t(\cdot \mid y)$ is the marginal at time $t$. A neural network $s_\phi(\mathbf{x}, y, t) \approx \nabla_{\mathbf{x}} \log p_t(\mathbf{x} \mid y)$ is trained via denoising score matching to approximate the score function:

$$\mathcal{L}_{\text{DSM}} = \mathbb{E}_{t, \mathbf{x}_0, \boldsymbol{\epsilon}} \left[ \lambda(t) \left\| s_\phi(\mathbf{x}_t, y, t) + \frac{\boldsymbol{\epsilon}}{\sigma(t)} \right\|^2 \right], \quad (3)$$

with $\mathbf{x}_t = \mathbf{x}_0 + \sigma(t)\, \boldsymbol{\epsilon}$ and $\boldsymbol{\epsilon} \sim \mathcal{N}(\mathbf{0}, \mathbf{I})$.

**Optimization-based Generation**  One of the most widely used optimization-based methods for 3D generation is

SDS (Poole et al., 2023). It repurposes a pretrained 2D score network $s_\phi$ (Rombach et al., 2022; Shi et al., 2023) to optimize 3D object parameters $\theta$ (e.g., NeRF weights). Let $I(\theta, v) = \text{Render}(\theta, v)$ be the image produced by a differentiable renderer parameterized by $\theta$ from viewpoint $v$. The SDS loss provides a gradient, $\nabla_\theta \mathcal{L}_{\text{SDS}}$, which steers $\theta$ over thousands of iterations to match the 2D diffusion prior.

$$\nabla_\theta \mathcal{L}_{\text{SDS}} \approx \mathbb{E}_{t,\epsilon}\left[w(t)\left(s_\phi(\mathbf{x}_t, y, t) + \tfrac{\epsilon}{\sigma(t)}\right)\frac{\partial I(\theta, v)}{\partial \theta}\right]. \quad (4)$$

This per-object optimization process, while slow, can produce high-fidelity assets. Our VLM3D reward ($r_{VLM}$) can be integrated as a powerful guidance signal within this iterative optimization framework.

**Feed-Forward Generation**    Feed-forward methods produce a 3D asset efficiently, comprising two major variants:

- Hybrid Feed-Forward: These methods first generate consistent multi-view 2D images via a diffusion model, and then apply a deterministic reconstruction algorithm to obtain the final 3D shape (Li et al., 2023b; Hong et al., 2023b; Xu et al., 2024).

- Native 3D Feed-Forward: SOTA models like Hunyuan3D (Hunyuan3D et al., 2025; Zhao et al., 2025) and CLAY (Zhang et al., 2024) are true 3D generative models that operate directly on a 3D latent representation (e.g., defined by an encoder $\mathcal{E}^*$ and decoder $\mathcal{D}^*$). In this case, a *single feed-forward generation* requires multiple internal iterative steps of the core network (e.g., a DiT) to solve the reverse SDE and sample the final 3D asset.

Crucially, this iterative *inference* process is distinct from the iterative *optimization* process of SDS (which runs for thousands of steps). As discussed in Sec. 2.2, while these efficient models often produce assets with sharp shapes and rich geometric details, their reliance on limited training data can lead to weaknesses in fine-grained semantic understanding and complex multi-object spatial reasoning. This makes them prime candidates for our test-time guidance module.

## 4. Method

We now describe our proposed framework, VLM3D, which provides a general, differentiable semantic and spatial critic from a pre-trained VLM. In Sec. 4.1 and Sec. 4.2, we detail the core of our framework: the definition of our differentiable VLM signal and the dual-query design that enables it to act as a semantic and spatial critic. Finally, in Sec. 4.3, we describe how this critic is integrated into two distinct paradigms: (1) optimization-based pipelines (Poole et al., 2023; Shi et al., 2023), and (2) SOTA native 3D feed-forward model (Hunyuan3D et al., 2025).

### 4.1. Vision-Language Model as an Explicit Semantic and Spatial Critic

We leverage a pre-trained VLM to provide a fully differentiable critic that measures both semantic fidelity and geometric consistency across multi-view renderings. Concretely, let the set of $N$ images rendered from our 3D representation parameters $\theta$ under viewpoints $v_i$

$$\mathcal{X} = \{x_i\}_{i=1}^N = \{I(\theta, v_1), \ldots, I(\theta, v_N)\} \quad (5)$$

and the user's text prompt $\mathbf{y}$ be the two inputs to the large VLM, we constrain the VLM function $Q(\cdot, \cdot)$, via carefully engineered queries, to output exclusively "Yes" or "No",

$$Q(\mathbf{y}, \mathcal{X}) \mapsto \{\text{Yes}, \text{No}\}. \quad (6)$$

Then we extract the final "Yes" and "No" logits of the VLM's binary-classification head, $z_{\text{yes}}$ and $z_{\text{no}}$. The corresponding probabilities are

$$P(\text{Yes} \mid \mathbf{y}, \mathcal{X}) = \frac{e^{z_{\text{yes}}}}{e^{z_{\text{yes}}} + e^{z_{\text{no}}}}, \quad P(\text{No} \mid \mathbf{y}, \mathcal{X}) = \frac{e^{z_{\text{no}}}}{e^{z_{\text{yes}}} + e^{z_{\text{no}}}}. \quad (7)$$

We define the VLM critic based on the log-odds:

$$r_{\text{VLM}} = \log P(\text{Yes} \mid \mathbf{y}, \mathcal{X}) - \log P(\text{No} \mid \mathbf{y}, \mathcal{X}) = z_{\text{yes}} - z_{\text{no}}. \quad (8)$$

The entire mapping $\theta \to \mathcal{X} \to r_{\text{VLM}}$ is differentiable, so we can backpropagate through the VLM to update $\theta$ directly.

### 4.2. VLM Dual-Query Design

Our full query $\mathbf{y}$ to the VLM is structured as follows:

*"Carefully evaluate the provided images, which show multiple views of a single 3D object. Does the underlying 3D object, considering all views together, meet all of the following criteria simultaneously?*
*1. Content Match: The object corresponds to the description: {text description}.*
*2. Geometric Quality: Based on all views combined, the object appears geometrically sound and consistent. There are no visible signs of major flaws such as multiple faces on one part (Janus-faced issue), broken surfaces, intersecting geometry, or highly unrealistic polygonal facets when considering the object from these different perspectives.*
*Strictly respond with only 'Yes' or 'No'."*

Our framework tasks the VLM to act as a dual-objective critic. It evaluates the multi-view renderings against this structured query comprising two distinct criteria: (1) Content Match (assessing semantic fidelity) and (2) Geometric Quality (assessing spatial coherence). This dual-query setup ensures that the critic in Eq. 8 captures both semantic alignment and spatial coherence.

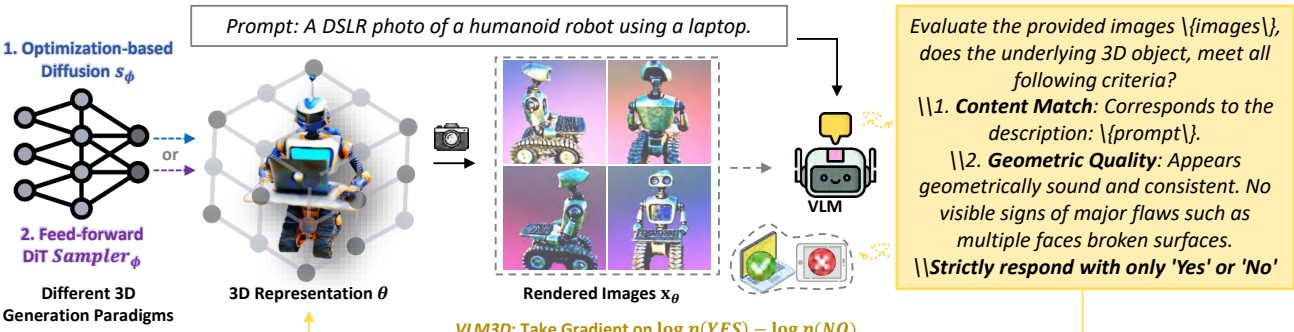

*Figure 2.* **Overview of VLM3D as a General Critic Framework.** Our core contribution, a differentiable dual-query $r_{\text{VLM}}$, acts as a versatile critic for 3D generation. It can be applied in two distinct paradigms: (1) As a Reward Objective: It is integrated into optimization-based pipelines (e.g., SDS (Poole et al., 2023)), replacing 2D priors with rich semantic and spatial reward. (2) As a Test-Time Guidance Module: It is used to guide the 3D assets sampling process of the native 3D models (e.g., Hunyuan3D (Hunyuan3D et al., 2025)), correcting their semantic and spatial errors.

## 4.3. Application Frameworks

Our general $r_{\text{VLM}}$ reward can be flexibly integrated into different 3D generation paradigms. We demonstrate its versatility in two primary frameworks.

### 4.3.1. REWARD FOR OPTIMIZATION-BASED PIPELINES

In the first framework, $r_{\text{VLM}}$ serves as the key reward objective within an optimization-based pipeline. The full training objective combines the VLM reward with the standard SDS loss (Poole et al., 2023), which provides texture and style priors. Specifically, we minimize:

$$\mathcal{L}_{\text{total}} = \mathcal{L}_{\text{SDS}} - \lambda_{\text{VLM}} r_{\text{VLM}}. \tag{9}$$

where $\lambda_{\text{VLM}}$ is the balancing factor.

We adopt a dynamic schedule for $\lambda_{\text{VLM}}$ during training. Initially, we set $\lambda_{\text{VLM}}$ high to enforce strong semantic and geometric constraints, ensuring the coarse shape aligns with the prompt. We then gradually decay $\lambda_{\text{VLM}}$ so that the SDS loss predominates, refining textures and fine details. This annealing schedule accelerates convergence and yields high-fidelity, semantically precise assets.

### 4.3.2. TEST-TIME GUIDANCE FOR FEED-FORWARD MODELS

In the second framework, $r_{\text{VLM}}$ is used as a test-time guidance signal, deeply integrated into the iterative sampling process of pre-trained, native 3D feed-forward models (like the DiT-based Hunyuan3D (Hunyuan3D et al., 2025)). At each step $t$ of the $T$-step denoising process (Sec. 3), our method computes the VLM guidance gradient $\nabla_{\mathbf{z}_t} r_{\text{VLM}}$ based on a differentiable rendering of the current prediction. This gradient is then used to modify the sampling direction, conceptually:

$$\mathbf{z}_{t-1} = \text{Sampler}(\mathbf{z}_t, t) + \lambda_{\text{TTG}} \nabla_{\mathbf{z}_t} r_{\text{VLM}}. \tag{10}$$

This in-process guidance acts as a powerful semantic and spatial critic, steering the generation away from incoherent paths before they solidify. As in (Laroche et al., 2024; Bai et al., 2024), this test-time guidance approach requires no model re-training.

## 5. Experiment

We now evaluate VLM3D's performance, demonstrating its effectiveness and generality across two major 3D generation paradigms. We first present our results integrating $r_{\text{VLM}}$ into optimization-based pipelines, followed by our results applying $r_{\text{VLM}}$ as a test-time guidance module to SOTA feed-forward models. We conclude with ablation studies. Additional implementation details and results are provided in the supplementary material.

### 5.1. VLM Reward for Optimization-based Pipeline

For this paradigm, we use MVDream (Shi et al., 2023) as our text-to-image diffusion backbone and Qwen2.5-VL 7B (Bai et al., 2025b) as the VLM reward backbone. The annealing schedule for $\lambda_{\text{VLM}}$ is detailed in Sec. 4.3.1. All experiments are run on a single NVIDIA A100 GPU with 2.2 hours per 3D asset.

**Setup** We evaluate on the comprehensive GPTEval3D benchmark (Wu et al., 2024b), which contains a diverse set of text prompts. Following this benchmark, we employ GPT-4o-mini to perform pairwise comparisons, calculating Elo ratings that emulate human judgments of text alignment, 3D plausibility, and texture-geometry coherence. We compare against representative score distillation-based methods: DreamFusion (Poole et al., 2023), DreamGaussian (Tang et al., 2023), ProlificDreamer (Wang et al., 2023), MV-Dream (Shi et al., 2023), DreamReward (Ye et al., 2024), and DreamDPO (Zhou et al., 2025).

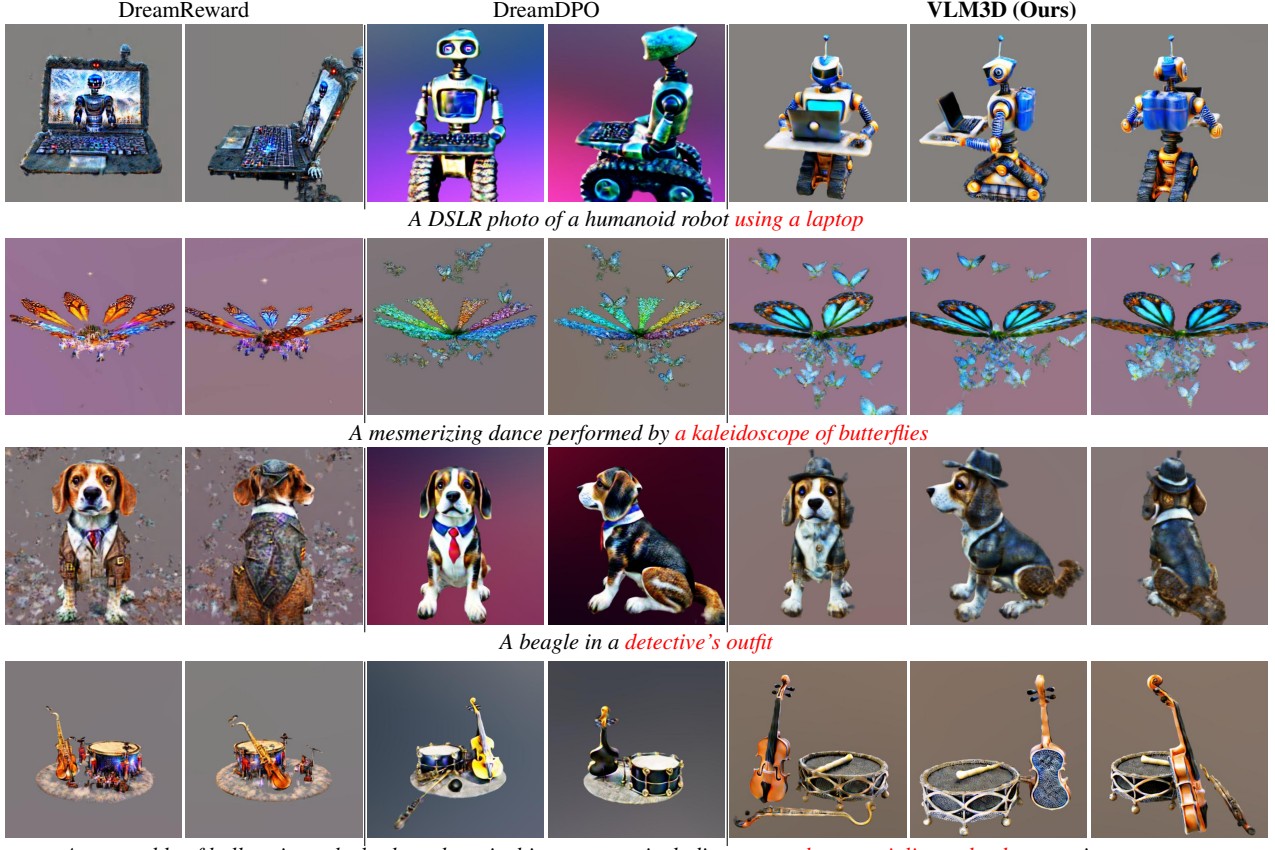

DreamReward  DreamDPO  **VLM3D (Ours)**

*A DSLR photo of a humanoid robot using a laptop*

*A mesmerizing dance performed by a kaleidoscope of butterflies*

*A beagle in a detective's outfit*

*An ensemble of hollow, irregularly shaped musical instruments, including a saxophone, a violin, and a drum resting on a stage*

*Figure 3.* **Comparison of VLM3D with optimization-based baselines.** VLM3D outperforms these methods (Ye et al., 2024; Zhou et al., 2025) in semantic fidelity while retaining high perceptual quality. Although baseline methods achieve good texture and detail—via differentiable preference rewards or non-differentiable optimization—they often miss fine-grained concepts (highlighted in red) that VLM3D captures accurately.

**Quantitative Evaluation on GPTEval3D** Table 1 reports performance of VLM3D against baselines on six metrics: Text–Asset Alignment, 3D Plausibility, Texture Details, Geometry Details, Texture–Geometry Coherence, and Overall Score. VLM3D—which integrates explicit VLM critic with SDS—outperforms all baselines on every metric. In particular, it achieves the top Text–Asset Alignment score, confirming that VLM critic substantially enhances semantic fidelity. It also leads in 3D Plausibility and perceptual quality (both texture and geometry). Overall, VLM3D's aggregate score surpasses the strongest baseline by a large margin, demonstrating the complementary benefits of combining SDS and VLM-based critic.

**Qualitative Comparison** Fig. 3 shows example 3D assets generated by current SOTA baselines (Ye et al., 2024; Zhou et al., 2025) for several prompts. Although these methods boost perceptual quality, they still lag behind in prompt fidelity. In contrast, VLM3D more effectively captures object interactions—rendering actions like "using a laptop"—and accurately models rare concepts such as "kaleidoscope" and "detective's outfit." It also excels at complex multi-object scenes, as shown by the jazz concert example with multiple

instruments, including "a saxophone, a violin, and a drum".

### 5.2. VLM Guidance for Feed-Forward Model

For this paradigm, we build upon the open-source pipeline of Hunyuan3D 2.0 (Hunyuan3D et al., 2025; Zhao et al., 2025). We apply our VLM3D reward (using the same Qwen2.5-VL 7B backbone (Bai et al., 2025a)) as a test-time guidance signal. This guidance is applied during the full 50 sampling steps of the Hunyuan3D shape generation pipeline (3D DiT (Peebles & Xie, 2023)), steering the sampling of 3D latents. All experiments are run on a single NVIDIA A100 GPU with 60 seconds per 3D asset.

**Setup** To evaluate this test-time guidance, we use a suite of standard metrics including CLIP-D (text-shape alignment), FID and CLIP-FID (realism), and a comprehensive Geometry score on all cases shown in Fig. 1 and 4. We compare our VLM critic-based test-time guidance results directly against the original outputs of Hunyuan3D 2.1 (Hunyuan3D et al., 2025) and CLAY (Zhang et al., 2024).

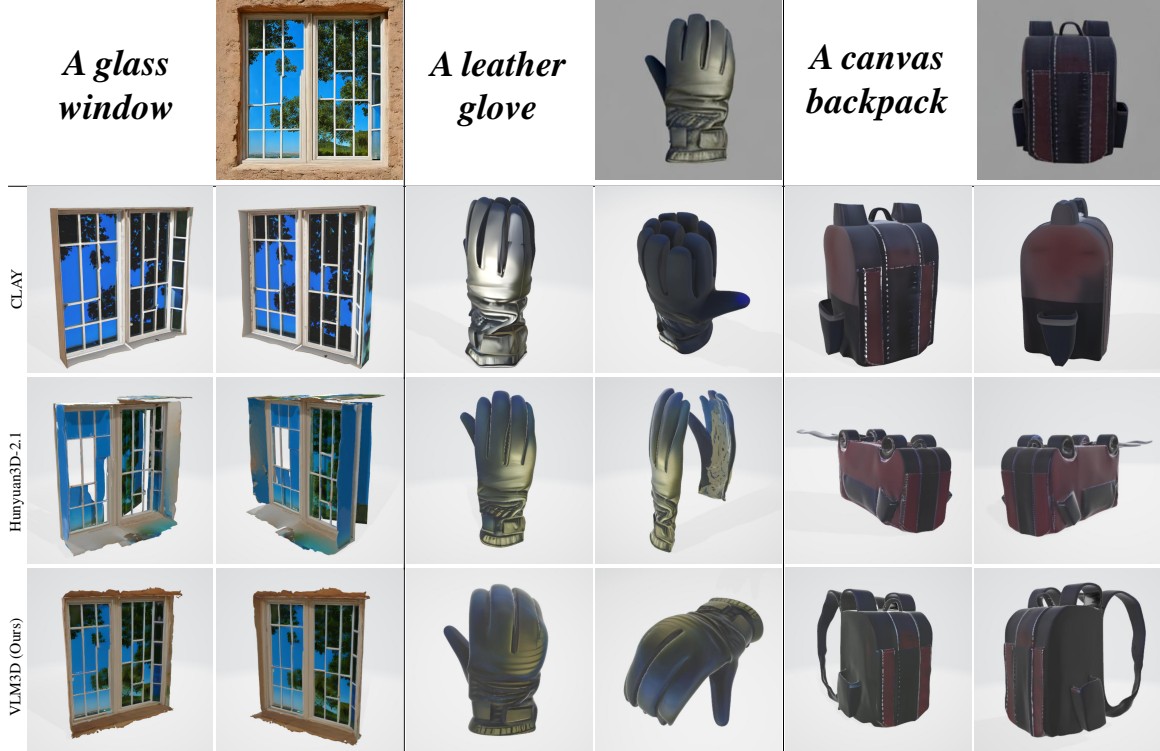

*Figure 4.* **Comparison of VLM3D with feed-forward baselines.** We present a qualitative comparison with SOTA native 3D models (Zhang et al., 2024; Hunyuan3D et al., 2025). Baselines exhibit significant failures, generating incomplete geometry, disconnected parts or distorted shapes. Our VLM3D, integrated as a test-time guidance module to the (Hunyuan3D et al., 2025) pipeline, successfully corrects these severe spatial faults and produces coherent 3D assets.

**Quantitative Evaluation** We present quantitative results in Table 2. The VLM-guided generation process yields significant improvements across the board. The substantial gain in CLIP-D demonstrates that our $r_{\text{VLM}}$ effectively steers the sampling process towards better text-semantic alignment. Furthermore, the improvement in the geometry scores confirms that our dual-query critic's "Geometric Quality" check (Sec. 4.2) is crucial for fixing the spatial issues inherent in the original feed-forward generation.

**Qualitative Comparison** As shown in Fig. 1 and 4, our test-time guidance demonstrates a profound impact on spatial intelligence and part-assembly, a key weakness of SOTA feed-forward models (Zhang et al., 2024; Hunyuan3D et al., 2025; Zhao et al., 2025). For the prompt "a Navy sailor" the baseline Hunyuan3D generates a nonsensical collection of floating, disconnected parts. In stark contrast, our VM3D-guided generation process, using the *exact same* feed-forward backbone, successfully assembles these components into a coherent object. Similarly, our VLM-based guidance correctly generates the full "leather glove" and the complete "glass window" with its frame, whereas the baselines produce incomplete or fractured geometry. This proves that our $r_{\text{VLM}}$ can serve as an effective spatial critic to guide the DiT sampler away from incoherent states.

### 5.3. More Analyses and Ablation Studies

In this section, we dissect VLM3D's behavior and quantify how key design choices affect 3D generation performance.

**Sensitivity to Prompt Perturbations** We assess VLM3D's semantic fidelity on prompts that differ by a single concept. As shown in Fig. 5, we modify a farmer scene by changing the clothing color and specifying "chopping through a tree trunk." VLM3D accurately reflects both the new color and the action, whereas MVDream fails to capture this detail action. Remarkably, VLM3D also exhibits superior spatial reasoning: when prompted to place a green apple "in" versus "beside" a plate, VLM3D honors each spatial relation, while MVDream fails. These results demonstrate VLM3D's strong semantic alignment and its sensitivity to even subtle prompt variations.

**Ablation: Geometric Query in VLM Prompt** We assess the effect of explicitly querying geometric consistency in the VLM prompt. Without this query, VLM3D exhibits classic multi-face artifacts on a cat prompt, and generates bicycles with floating parts and fractured surfaces (Fig. 7). Including the geometric query substantially mitigates these errors, confirming that explicit geometric guidance is critical.

*Table 1.* **Quantitative Results on 110 Prompts from the GPTEval3D Benchmark (Wu et al., 2024b).** We compute all six GPTEval3D metrics—text alignment, 3D plausibility, texture–geometry coherence, geometry details, texture details, and overall score—to comprehensively evaluate 3D generation quality. VLM3D achieves the highest score on every metric, demonstrating its superior performance.

| Method | Prompts from GPTEval3D | | | | | |
| --- | --- | --- | --- | --- | --- | --- |
| | Alignment↑ | Plausibility↑ | T-G Coherency.↑ | Geo Details↑ | Tex Details↑ | Overall↑ |
| DreamFusion(Poole et al., 2023) | 1000.0 | 1000.0 | 1000.0 | 1000.0 | 1000.0 | 1000.0 |
| DreamGaussian(Tang et al., 2023) | 1100.6 | 953.6 | 1158.6 | 1126.2 | 1130.8 | 951.4 |
| Latent-NeRF(Metzer et al., 2022) | 1222.3 | 1144.8 | 1156.7 | 1180.5 | 1160.8 | 1178.7 |
| ProlificDreamer(Wang et al., 2023) | 1261.8 | 1058.7 | 1152.0 | 1246.4 | 1180.6 | 1012.5 |
| MVDream(Shi et al., 2023) | 1270.5 | 1147.5 | 1250.6 | 1324.9 | 1255.5 | 1097.7 |
| DreamReward(Ye et al., 2024) | 1287.5 | 1195.0 | 1254.4 | 1295.5 | 1261.6 | 1193.3 |
| DreamDPO(Zhou et al., 2025) | 1298.9 | 1171.9 | 1276.4 | 1373.2 | 1296.9 | 1203.1 |
| VLM3D (Ours) | **1365.5** | **1293.7** | **1365.4** | **1419.0** | **1368.7** | **1268.6** |

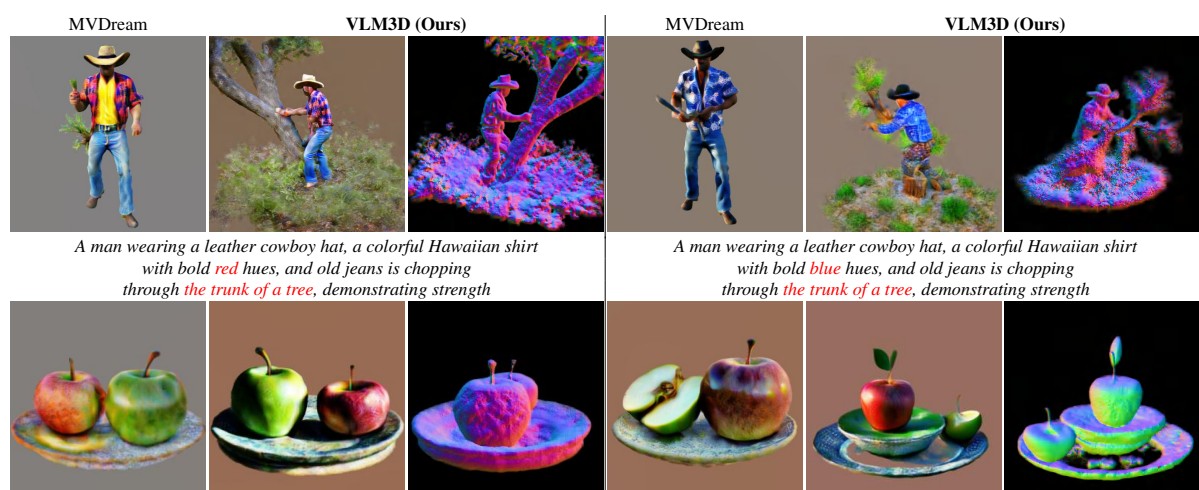

*Figure 5.* **Sensitivity Analysis to Text Perturbations.** We compare VLM3D and MVDream on pairs of prompts that differ by a single concept (highlighted in red). VLM3D accurately changes clothing color (first row), and updates spatial relations (second row), demonstrating its better semantic understanding than baselines.

*Table 2.* **Quantitative Results for feed-forward pipelines on 24 prompts.** VLM3D boosts both semantic and geometric quality.

| Method | CLIP-D ↓ | FID ↓ | CLIP-FID ↓ | Geo. ↓ |
| --- | --- | --- | --- | --- |
| CLAY (Zhang et al., 2024) | 0.22 | 310.3 | 53.11 | 0.63 |
| Hunyuan3D (Zhao et al., 2025) | 0.23 | 338.6 | 54.01 | 0.58 |
| VLM3D (Ours) | **0.19** | **274.9** | **45.79** | **0.49** |

**Ablation: Multi-View vs. Single-View Inputs** We further assess whether querying the VLM with multiple views is necessary for spatial coherence. When the VLM receives only a single rendered image instead of the full view set, VLM3D again suffers from Janus artifacts (Fig. 7). Multiview inputs are critical to enforcing 3D consistency.

## 6. Conclusion

In this paper, we presented VLM3D, a novel framework that establishes large VLMs as general, differentiable semantic-and-spatial critic for text-to-3D generation. By introducing a dual-query reward—one for content fidelity and one for geometric constraints—and an end-to-end differentiable pipeline that backpropagates VLM log-odds, VLM3D achieves precise prompt alignment, mitigates view-consistency artifacts, and produces high-fidelity textures and geometry. We validate this in both SDS and native 3D generation paradigms.

**Limitations and Future Directions** Despite these advances, our VLM reward formulation can still struggle with very long or highly detailed prompts. For instance (Fig. 1), when tasked with generating the renowned "Embracing Peace" statue, VLM3D correctly reproduces the kissing pose and dual figures—outperforming MVDream, which omits the nurse entirely—but still misses finer details such as the nurse's lifted leg and outstretched arm. Interestingly, a standalone VLM can readily describe these details from a photograph, indicating that our current reward formulation does not fully leverage the VLM's rich visual reasoning.

Building on these insights, we see several promising paths

forward. First, disentangling semantic and geometric feedback—perhaps via separate VLM heads for content and geometry queries—might provide finer control over prompt fidelity and 3D consistency. Second, advanced query engineering, such as hierarchical or detail-focused cues, could ensure the VLM reward faithfully captures nuanced attributes and fine-grained descriptions.

## Impact Statement

This paper aims to advance text-to-3D generation by improving the semantic alignment and geometric consistency of generated 3D assets. This paper presents work to advance the field of machine learning. There are many potential societal consequences of our work, none of which we feel must be specifically highlighted here.

## Acknowledgement

This work was supported by the National Natural Science Foundation of China (62371007, 32450631), the National Key Research and Development Program of China (2024YFC3406400), the Shanghai Municipal Science and Technology Major Project (2025SHZDZX026D03) and the projects of Beijing Science and Technology Program(Z251100008125028). Additionally, we acknowledge the support from the Biomedical Computing Platform of National Biomedical Imaging Center, Peking University.

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

# A. Implementation Details

In this section, we provide detailed implementations of our VLM3D framework across two generation paradigms. Section A.1 details the optimization-based pipeline, and Section A.2 introduces the details for the feed-forward pipeline. Finally, implementation details for the baseline methods are provided in Section A.3. To facilitate evaluation and reproducibility, we provide a compressed package containing the source code and videos in the attachment. We will also publish it in a public repository.

## A.1. VLM3D (Optimization-based Pipeline)

**Pseudo-code for VLM3D** A detailed pseudo-code for VLM3D is presented in algorithm 1.

**Hyperparameters** The optimization of the 3D representations, NeRFs in this work, typically takes $10,000$ to $15,000$ update steps, with a learning rate set to $1 \times 10^{-3}$. In our text-to-image diffusion model, we employ a classifier-free guidance (CFG) loss with a scale of $50$. While common sparsity and opacity regularizations are omitted, we retain orientation regularization, annealing its weight from $10$ to $1000$ over the initial $5000$ training iterations.

Among the hyperparameters, the weight assigned to the Vision-Language Model (VLM) reward is particularly critical. As detailed in the manuscript, this weight is generally annealed from a high initial value to a lower one during training; here, we elaborate on this annealing strategy. For challenging prompts—those characterized by significant length, multiple objects, complex spatial relationships, or infrequent terminology—we set the initial VLM reward weight to a value within the range of $[300, 800]$. This higher initial weight empowers the VLM to comprehensively interpret these intricate prompts, thereby establishing a robust foundation for generation. Conversely, for simpler prompts that are readily processed by standard Stable Diffusion-based methods, such a high initial weight is not strictly necessary, although it remains effective. Overall, this annealing schedule for the VLM reward weight consistently yields 3D assets with high-quality geometry and strong prompt alignment, steadily outperforming SDS-based methods, especially on more demanding prompts.

**Backbones of Diffusion Models and VLMs** For our text-to-image diffusion backbone, we evaluated two models: Stable Diffusion v2.1 (Rombach et al., 2022) and the fine-tuned Stable Diffusion v2.1-base-4view (Shi et al., 2023). The latter demonstrated superior robustness and capability in 3D generation tasks, and was therefore adopted as our primary diffusion model. Similarly, for the Vision-Language Model (VLM) component, we assessed three prominent open-source options: Qwen2.5-VL (7B) (Bai et al., 2025b), PaliGemma (3B) (Beyer et al., 2024), and IDEFICS. Our preliminary analysis indicated that Qwen2.5-VL (7B) delivered the most compelling performance, leading to its selection as our VLM backbone.

**Differentiable Image Processors for VLMs** VLM3D shares foundational principles with Reinforcement Learning from Human Feedback (RLHF) (Christiano et al., 2017; Ouyang et al., 2022), a paradigm widely recognized for its power in aligning models with reward models. This perceived superiority of RLHF typically stems from its use of explicit reward modeling and its inherent capacity for exploration during the differentiable learning process.

Despite its potential, the practical application of RLHF or similar gradient-based reward optimization for VLMs encounters significant obstacles. Even among prominent open-source VLMs, a common architectural design involves various image preprocessors that detach the gradient flow through the visual components. *We identify that a key challenge in enabling end-to-end differentiability through VLMs is that their image preprocessors typically detach gradients to accommodate diverse input formats, often by converting image data to NumPy arrays for intermediate processing steps.* This detachment cuts off the pathway for backpropagating reward signals derived from the VLM, a critical requirement for end-to-end gradient-based optimization techniques that leverage VLM feedback.

A core technical contribution of this work is the establishment of a fully differentiable pathway through the VLM, which is crucial for our VLM3D framework. To overcome the detachment issue in the image processors, we have re-engineered this module by redesigning its internal operations to exclusively utilize Torch tensors, thereby maintaining an uninterrupted gradient flow. By ensuring continuous gradient flow, we enable an end-to-end differentiable forward process. This crucial modification allows the VLM's rich semantic and spatial understanding to be translated into well-defined, differentiable reward signals, directly informing and refining the generation of 3D assets. We have attached the re-engineered preprocessor code in the supplementary material.

---

**Algorithm 1** Pseudo-code for VLM3D (Optimization-based Pipeline)

---

0: **Input:** Text-to-image diffusion model $s_\phi$, VLM $Q(\cdot, \cdot)$, Text prompt $y$, Number of views $N$
0: **Input:** Learning rate $\eta$ for 3D representation $\theta$, Annealing schedule $\lambda_{VLM}$
0:
0: **Initialization:** Random A 3D representation (e.g., NeRF) parameterized by $\theta_0$
0:
0: **while** not converged **do**
0:     Sample $N$ camera viewpoints $\{v_i\}_{i=1}^N$
0:     Render $N$ images $\mathcal{X} = \{x_i\}_{i=1}^N = \{I(\theta, v_i)\}_{i=1}^N$
0:
0:     **VLM Reward**
0:         Query VLM: $Q(y, \mathcal{X}) \mapsto \{\text{Yes}, \text{No}\}$ using the designed prompt
0:         Extract "Yes" ($z_{yes}$) and "No" ($z_{no}$) logits from VLM
0:         Calculate VLM reward: $r_{VLM} = z_{yes} - z_{no}$
0:
0:     **SDS Loss**
0:         Sample timestep $t \sim \text{Uniform}(0, T)$ and noise $\epsilon \sim \mathcal{N}(0, I)$
0:         Compute noisy image $x_t = x_0 + \sigma(t)\epsilon$
0:         Estimate score $s_\phi(x_t, y, t)$
0:         SDS gradient: $\nabla_\theta \mathcal{L}_{SDS} \approx w(t)(s_\phi(x_t, y, t) + \frac{\epsilon}{\sigma(t)})\frac{\partial I(\theta, v_i)}{\partial \theta}$
0:
0:     **Total Loss and Parameter Update**
0:         Retrieve current VLM weight $\lambda_{VLM}$ from annealing schedule
0:         Compute total loss gradient: $\nabla_\theta \mathcal{L}_{total} = \nabla_\theta \mathcal{L}_{SDS} - \lambda_{VLM} \nabla_\theta r_{VLM}$
0:         Update parameters: $\theta \leftarrow \theta - \eta \nabla_\theta \mathcal{L}_{total}$.
0: **end while**=0

---

## A.2. VLM3D (Feed-Forward Pipeline)

Unlike the optimization-based paradigm which updates 3D parameters iteratively from scratch, our Feed-Forward framework operates as a test-time guidance (TTG) module. It intervenes in the sampling process of a pre-trained native 3D diffusion model without requiring model re-training.

**Guidance Mechanism.**   Since most state-of-the-art feed-forward 3D models are specialized for Image-to-3D generation, our pipeline adopts a two-stage approach. First, we generate a high-quality reference image $I_{\text{ref}}$ from the input text prompt $y$ using MVDream (Shi et al., 2023). This image $I_{\text{ref}}$ then serves as the condition for the feed-forward model (e.g., Hunyuan3D-2.1 (Hunyuan3D et al., 2025)). To enable gradient-based guidance during the sampling process, we implement a fully differentiable rendering pipeline utilizing the *nvdiff_rasterizer* module from the `nvdiffrast` library. In each denoising step $t$, the intermediate 3D latent is decoded and rendered into multi-view images $\mathcal{X}$ via this differentiable pipeline. The VLM critic then evaluates $\mathcal{X}$ against the original text prompt $y$ to compute the reward $r_{\text{VLM}}$. Finally, the guidance gradient $\nabla_{z_t} r_{\text{VLM}}$ is backpropagated through the differentiable rasterizer and the 3D decoder to update the latent $z_t$.

**Hyperparameters.**   The sampling process consists of $T = 50$ steps. We apply VLM guidance with a time-dependent weight $\lambda(t)$ using a linear decay schedule: the guidance scale starts at $\lambda_{\text{start}} = 10$ at $t = T$ and decays to $\lambda_{\text{end}} = 0$ at $t = 0$. This ensures that VLM3D corrects major semantic and structural errors early on while preserving fine textures generated by the native model. The rendering resolution is set to $512 \times 512$.

**Pseudo-code.**   The inference process with VLM3D test-time guidance is detailed in Algorithm 2.

## A.3. Baselines

**MVDream (Shi et al., 2023)**   For the MVDream baseline, we utilize the official codebase provided by (Shi et al., 2023). The text-to-image diffusion model employed is the fine-tuned Stable Diffusion v2.1-base-4view, also introduced in (Shi

---

**Algorithm 2** Pseudo-code for VLM3D (Feed-Forward-based Pipeline)

---

0: **Input:** Pre-trained 3D diffusion model $\epsilon_\phi$, 3D Decoder $\mathcal{D}^*$, Differentiable Renderer $\mathcal{R}$
0: **Input:** VLM $Q(\cdot, \cdot)$, Text prompt $y$, Total timesteps $T$, Guidance schedule $\{\lambda_t\}_{t=1}^T$
0:
0: **Initialization:** Sample Gaussian noise $z_T \sim \mathcal{N}(0, I)$
0:
0: **for** $t = T$ **to** $1$ **do**
0:     **Standard Diffusion Prediction**
0:         Estimate velocity/noise: $v_{\text{pred}} = \epsilon_\phi(z_t, t, y)$
0:         Estimate clean latent: $\hat{z}_0 \approx z_t - \sigma_t v_{\text{pred}}$
0:
0:     **Differentiable Evaluation Pipeline**
0:         Decode 3D representation: $\Theta = \mathcal{D}^*(\hat{z}_0)$
0:         Sample viewpoints $\{v_i\}_{i=1}^N$ and render: $\mathcal{X} = \mathcal{R}(\Theta, \{v_i\})$
0:
0:     **VLM Critic**
0:         Query VLM: $Q(y, \mathcal{X}) \mapsto \{\text{Yes}, \text{No}\}$
0:         Extract "Yes" ($z_{yes}$) and "No" ($z_{no}$) logits from VLM
0:         Compute Reward: $r_{VLM} = z_{yes} - z_{no}$
0:
0:     **Guidance and Update**
0:         Calculate gradient: $g = \nabla_{z_t} r_{VLM}$ (via backprop through $\mathcal{R}$ and $\mathcal{D}^*$)
0:         Retrieve current guidance scale $\lambda_t$
0:         Modify prediction: $\hat{v}_{\text{guided}} = v_{\text{pred}} - \lambda_t \cdot g$
0:         Update latent: $z_{t-1} \leftarrow \text{SamplerStep}(z_t, \hat{v}_{\text{guided}}, t)$
0: **end for**
0:
0: **Output:** Final 3D Asset $\mathcal{D}^*(z_0)$ =0

---

et al., 2023). We adhere to the original hyperparameter configurations.

**DreamReward (Ye et al., 2024)** Our implementation of DreamReward (Ye et al., 2024) is based on the official source code. We employ Stable Diffusion v2.1 (Rombach et al., 2022) as the text-to-image diffusion backbone, complemented by the official Reward3D Scorer [1] serving as the 3D reward model. All hyperparameters are kept consistent with the original implementation.

**DreamDPO (Zhou et al., 2025)** Considering that the official code for DreamDPO (Zhou et al., 2025) has not yet been publicly released, we directly report the results presented in the original paper for comparative analysis.

**CLAY (Zhang et al., 2024)** We utilize the official implementation of CLAY (Zhang et al., 2024). Specifically, we use the online *Rodin Gen-1* demo for generation, adhering to the official default settings.

**Hunyuan3D-2.1 (Hunyuan3D et al., 2025)** We employ the official open-source repository of Hunyuan3D-2.1 (Hunyuan3D et al., 2025). All experiments are conducted using the provided checkpoints and default configuration.

**Others** For results of other established methods, including DreamFusion (Poole et al., 2023), DreamGaussian (Tang et al., 2023), Latent-NeRF (Metzer et al., 2022), Magic3D (Lin et al., 2023), and ProlificDreamer (Wang et al., 2023), our experiments are primarily derived from the threestudio project [2]. Some results were also sourced from the GPTEval3D repository [3].

---

[1] The Reward3D weights can be found at https://huggingface.co/yejunliang23/Reward3D
[2] The threestudio can be found at https://github.com/threestudio-project/threestudio
[3] The benchmark GPTEval3D can be found at https://github.com/3DTopia/GPTEval3D

---

**Example Prompt A (Focus: Content Alignment Only)**

Carefully evaluate the provided images, which show multiple views of a single 3D object. Does the underlying 3D object, considering all views together, correspond to the description: '`[text_description_here]`'?
Strictly respond with only 'Yes' or 'No'.

---

**Example Prompt B (Focus: Simplified Criteria)**

Carefully evaluate the 3D object shown in the multiple input image views based on the following criteria:
1. **Content Match:** Does the object strongly match the text description: `[text_description_here]`?
2. **Visual Plausibility:** Does the object appear visually coherent and plausible across all views, without major jarring artifacts or inconsistencies?
Considering both criteria, answer 'Yes' if both are reasonably met. Otherwise, answer 'No'.

---

## B. Details of VLM prompt design

The design of the prompt provided to the Vision-Language Model (VLM) is a critical factor influencing the quality and relevance of the VLM reward for 3D generation. To identify the most effective prompt for our VLM3D framework, we experimented with various prompt formulations. Below, we present three representative examples from our exploration, including our finally selected prompt, and discuss the insights behind our choice.

**Discussion of Prompt Selection** Our empirical evaluations demonstrated that the "Selected VLM Prompt (Our Optimal Formulation)" yielded the best results for 3D generation. This prompt's effectiveness stems from several key attributes when compared to other variants, such as "Example Prompt A" and "Example Prompt B".

The chosen prompt excels due to its comprehensive yet clear criteria. It explicitly requires the VLM to assess both Content Match and Geometric Quality simultaneously, considering all views collectively. This dual-query structure is crucial, as focusing solely on content alignment, like in "Example Prompt A," often leads to 3D assets that match the text semantically but suffer from significant geometric flaws. The inclusion of an explicit geometric quality check, as detailed in our selected prompt and supported by ablation studies (see Fig. 7 and Fig. 6), is critical for ensuring 3D consistency and plausibility.

Furthermore, the "Geometric Quality" criterion in our selected prompt is highly specific, enumerating common failure modes like "multiple faces on one part, broken surfaces, intersecting geometry, or highly unrealistic polygonal facets." This level of detail provides clearer guidance to the VLM compared to more abstract phrasing. For instance, "Example Prompt B" uses the term "Visual Plausibility" and asks if the criteria are "reasonably met." While aiming for a similar goal, "Visual Plausibility" can be more subjective and may not as effectively penalize specific 3D inconsistencies as the detailed "Geometric Quality" checklist. The explicit instruction to consider the object "from these different perspectives" also reinforces the need for multi-view consistency.

Finally, the unambiguous instruction to "Strictly respond with only 'Yes' or 'No'" ensures a clean, binary signal for reward calculation, simplifying the integration of VLM feedback into our optimization pipeline. In contrast, prompts that might give less constrained responses could complicate the derivation of a differentiable reward.

The balance of comprehensiveness, specificity in defining undesirable artifacts, and a clear output format made our selected prompt the most robust and effective choice among the candidates evaluated.

## C. More Analysis

**Computational overhead.** We profile VLM3D and representative baselines on a single A100 GPU. As shown in Table 3, VLM3D is competitive with optimization-based methods: its 2.2-hour runtime is close to DreamDPO and much faster than ProlificDreamer. For the feed-forward setting, the VLM stage adds about 60 seconds of overhead. The memory cost mainly

> **Selected VLM Prompt (Our Optimal Formulation)**
>
> Carefully evaluate the provided images, which show multiple views of a single 3D object. Does the underlying 3D object, considering all views together, meet all of the following criteria simultaneously?
> 1. **Content Match:** The object corresponds to the description: `[text_description_here]`.
> 2. **Geometric Quality:** Based on all views combined, the object appears geometrically sound and consistent. There are no visible signs of major flaws such as multiple faces on one part (Janus-faced issue), broken surfaces, intersecting geometry, or highly unrealistic polygonal facets when considering the object from these different perspectives.
>
> Strictly respond with only 'Yes' or 'No'.

*Table 3.* Runtime and memory comparison on a single A100 GPU.

| Method | Setting | Time | Memory |
|---|---|---|---|
| DreamFusion | Optimization | 1.5 h | 12 GB |
| MVDream | Optimization | 1.0 h | 14 GB |
| ProlificDreamer | Optimization | 10.5 h | 27 GB |
| DreamReward | Optimization | 1.0 h | 22 GB |
| DreamDPO | Optimization | 2.0 h | 24 GB |
| VLM3D | Optimization | 2.2 h (SDS 55% + VLM 45%) | 49 GB |
| Hunyuan3D | Feed-forward | 60 s | 18 GB |
| VLM3D | Feed-forward | 120 s (base 60 s + VLM 60 s) | 53 GB |

comes from the VLM and can be reduced by quantization or a smaller backbone such as PaliGemma-3B.

**Semantic and geometric disentanglement.** To study whether the semantic and geometric queries provide complementary signals, we compare three prompt designs: the full dual-query prompt, a content-only prompt, and a geometry-only prompt. As shown in Table 4, all variants maintain stable gradients, while the dual-query design achieves the highest final reward. This supports our design choice that semantic correctness and geometric plausibility should be queried jointly but explicitly.

**Guidance scheduling.** Early VLM guidance is important because the coarse 3D structure is determined early in optimization; after convergence, the diffusion prior mainly refines surface details and textures, making global structural correction difficult. We therefore use a decaying VLM guidance schedule. Table 5 shows that decay-based schedules outperform the reverse schedule, which delays strong VLM guidance until it is less effective.

**Geometric metrics.** Our Geometry Score is a deterministic, non-learned metric defined as the equal-weighted average of Chamfer Distance, Hausdorff Distance, and symmetric Point-to-Mesh Distance between normalized meshes. It only uses Euclidean distances in 3D space and is independent of any VLM. We further evaluate using two external 3D-native metrics, ULIP (Xue et al., 2023) and Uni3D (Zhou et al., 2024), which directly process point clouds rather than 2D renderings. As shown in Table 6, VLM3D consistently improves both text- and image-conditioned 3D alignment.

**VLM-only optimization.** To verify that the VLM gradient provides a meaningful optimization direction rather than a noisy auxiliary signal, we evaluate a VLM-only variant without SDS. As shown in Table 7, VLM-only optimization already produces text-aligned 3D content, while the full VLM3D achieves the best CLIP score by combining VLM guidance with SDS-based appearance refinement.

## D. Additional Results

More visual examples are provided in Figure 8, Figure 9, Figure 10 and Figure 11. These results show that our method creates high-quality and human-preferred 3D assets. Additionally, these assets more accurately match the given text descriptions and display finer details in their shape and surface textures.

*Table 4.* Prompt ablation. We report gradient norm and VLM reward during optimization.

| Metric | Step | Dual-query | Content-only | Geometry-only |
|---|---|---|---|---|
| Gradient norm | 1000 | $8.28\times10^{-1}$ | $8.24\times10^{-1}$ | $8.87\times10^{-1}$ |
| Gradient norm | 5000 | $4.05\times10^{-1}$ | $7.21\times10^{-1}$ | $6.87\times10^{-1}$ |
| Gradient norm | 9000 | $2.84\times10^{-1}$ | $3.86\times10^{-1}$ | $2.90\times10^{-1}$ |
| Reward | 1000 | $-0.0469$ | $-0.1188$ | $-1.1589$ |
| Reward | 5000 | $3.1875$ | $1.9531$ | $1.8492$ |
| Reward | 9000 | $4.2500$ | $2.8619$ | $3.7718$ |

*Table 5.* Ablation on VLM guidance scheduling.

| Schedule | Avg. Reward ↑ |
|---|---|
| Constant | 3.02 |
| Reverse linear | 3.31 |
| Cosine decay | 4.18 |
| Linear decay | 4.44 |

*Table 6.* External 3D-native metric evaluation.

| Metric | Hunyuan3D | VLM3D |
|---|---|---|
| ULIP-T ↑ | 0.03488 | 0.03544 |
| ULIP-I ↑ | -0.00021 | 0.00450 |
| Uni3D-T ↑ | 0.14646 | 0.20110 |
| Uni3D-I ↑ | 0.19726 | 0.27607 |

*Table 7.* VLM-only quantitative ablation. CLIP score is computed as the cosine similarity between rendered views and the text prompt.

| Setting | CLIP Score ↑ |
|---|---|
| SDS-only, MVDream | 0.290 |
| VLM-only, no SDS | 0.271 |
| VLM3D, ours | 0.317 |

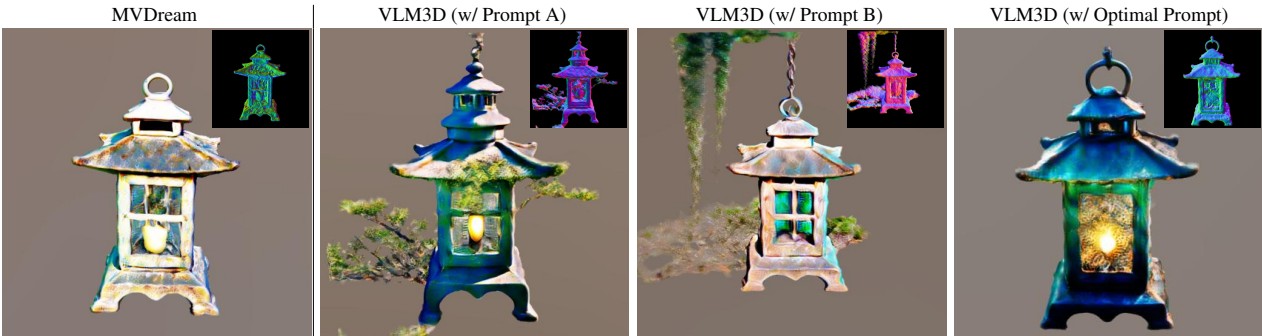

*An old, layered, asymmetrical lantern, with a patina copper finish and translucent panes that flicker with bioluminescent light from cultured algae within*

*Figure 6.* **Effect of different VLM prompt designs.**

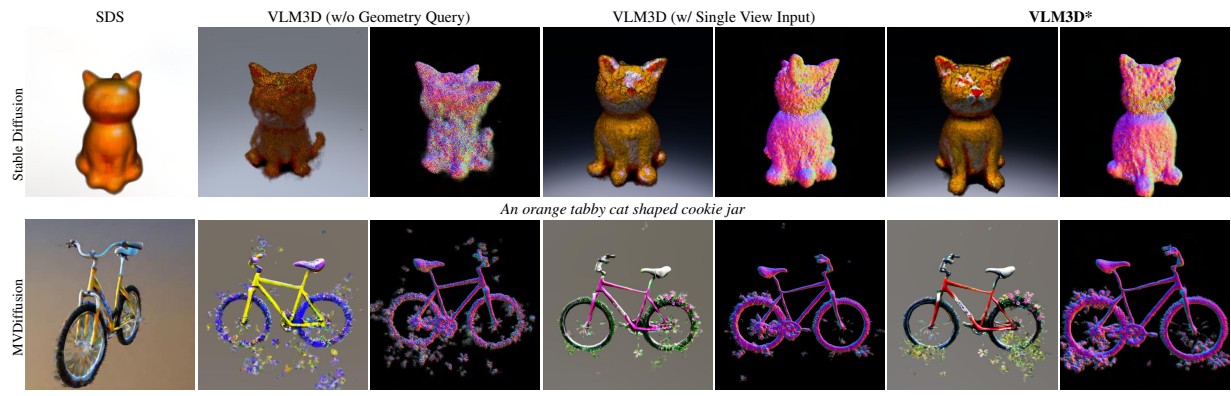

*Figure 7.* **Ablation of Geometric Query and Multi-View Input.** We assess the impact of (a) removing the explicit geometry-consistency query from the VLM prompt and (b) using a single view instead of multi-view images. Omitting either component degrades 3D quality—leading to Janus-face artifacts, floating parts, and fractured surfaces. Each row uses a different diffusion backbone: the top employs Stable Diffusion 2-1 (Rombach et al., 2022), while the bottom uses MVDream (Shi et al., 2023).

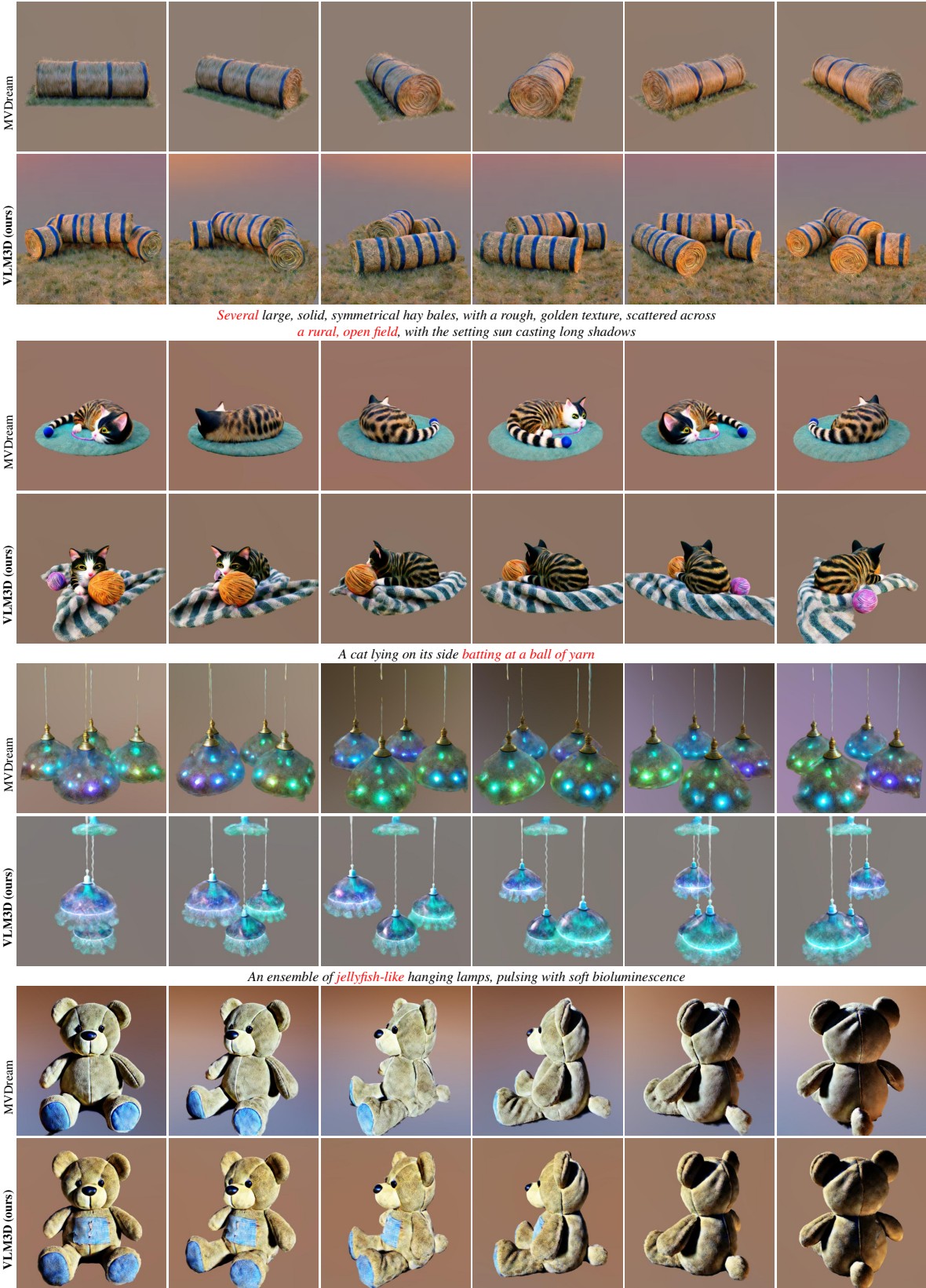

*Several large, solid, symmetrical hay bales, with a rough, golden texture, scattered across a rural, open field, with the setting sun casting long shadows*

*A cat lying on its side batting at a ball of yarn*

*An ensemble of jellyfish-like hanging lamps, pulsing with soft bioluminescence*

*A plush teddy bear, sitting alone with a slight tear in its seam*

*Figure 8.* **Additional results generated by our optimization-based VLM3D.**

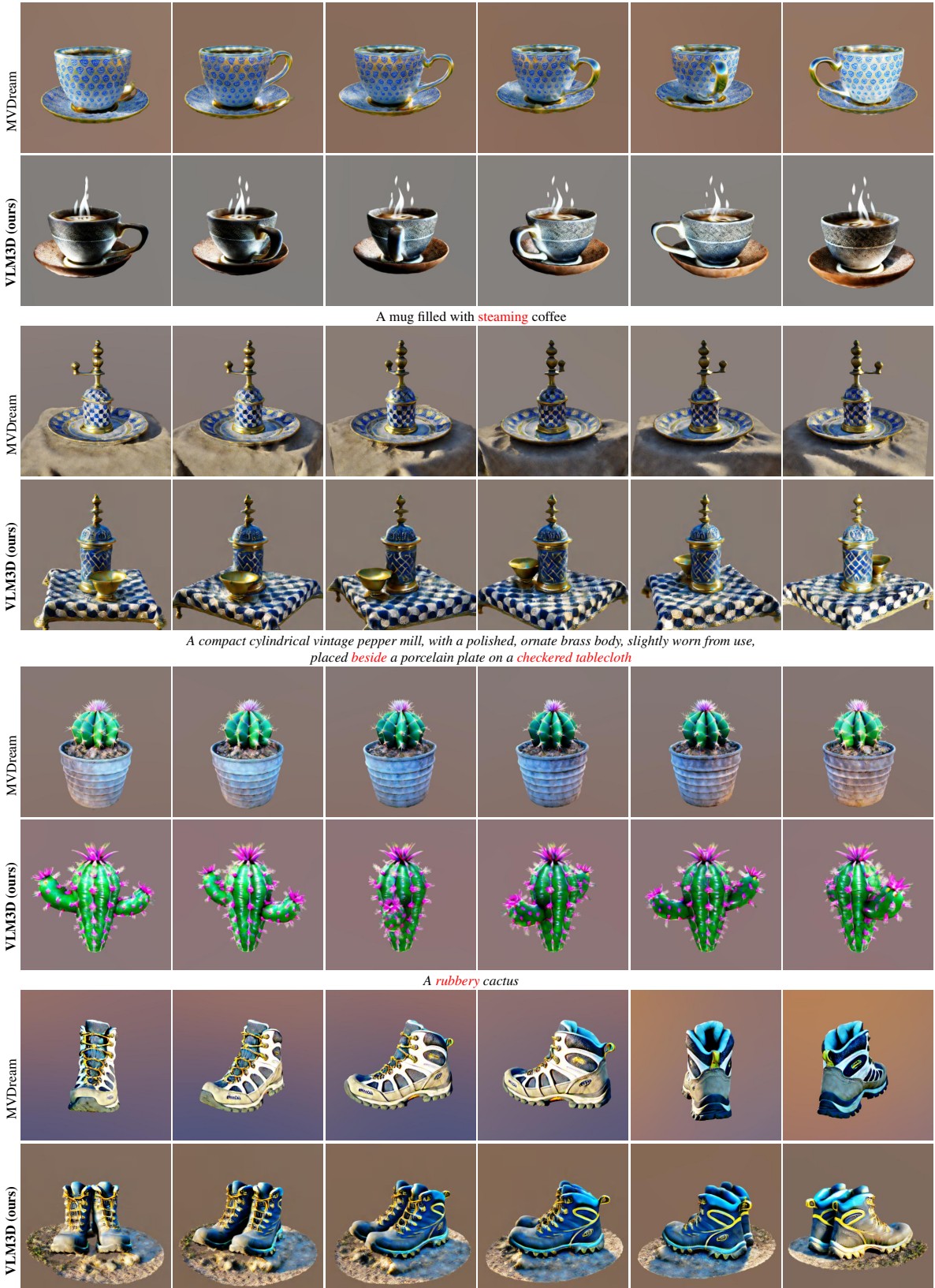

*Figure 9.* **Additional results generated by our optimization-based VLM3D.**

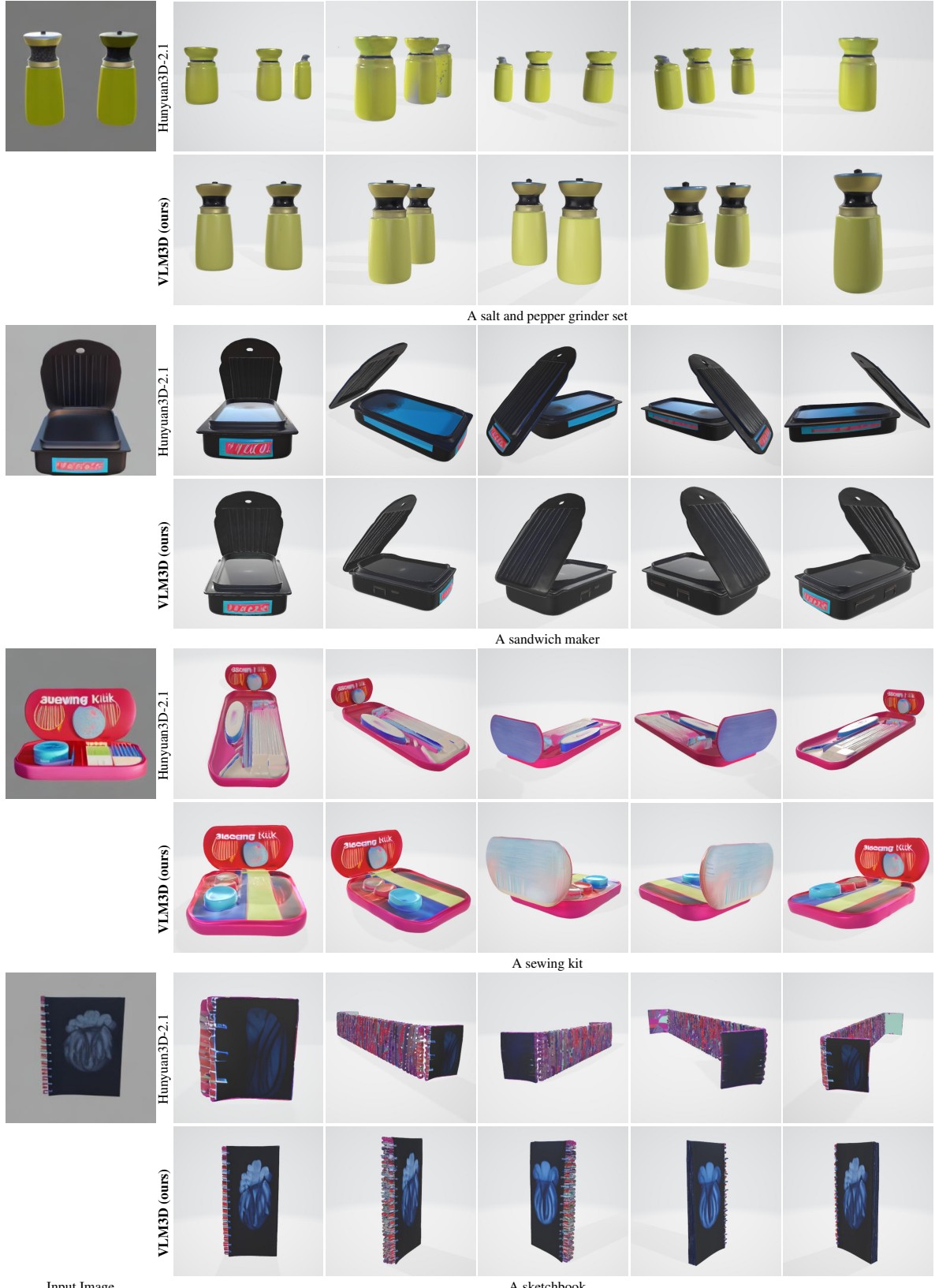

*Figure 10.* **Additional results generated by our feed-forward-based VLM3D.**

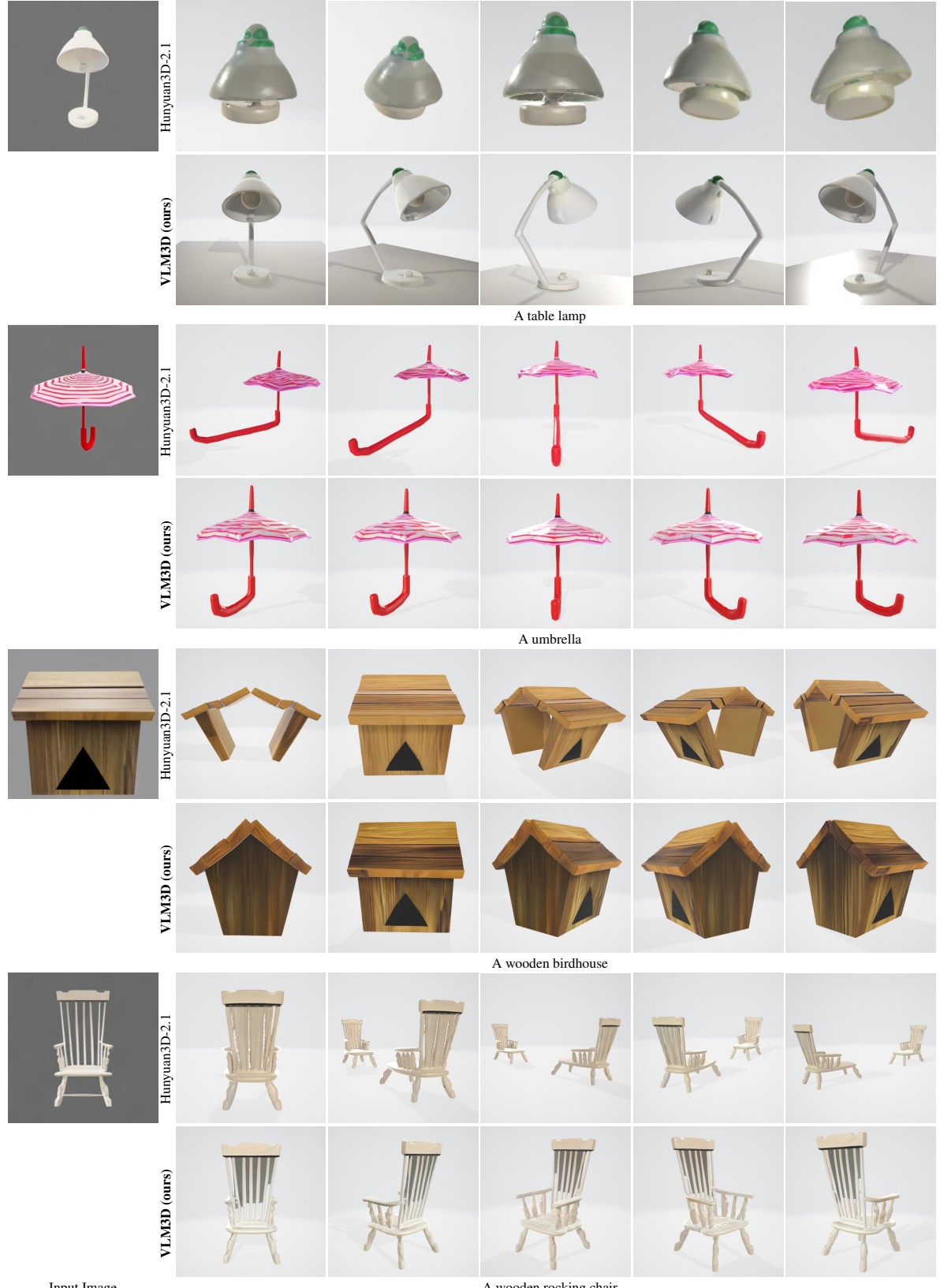

*Figure 11.* **Additional results generated by our feed-forward-based VLM3D.**

