# OpenReview forum: "Let Language Constrain Geometry: Vision–Language Models as Semantic and Spatial Critics for 3D Generation"
_ICML.cc/2026/Conference — ICML 2026 regular_

### Official Review · Reviewer_rG78 · 2026-02-28

**Soundness:** 4
**Presentation:** 3
**Significance:** 3
**Originality:** 3
**Overall Recommendation:** 5
**Confidence:** 4

**Summary:**

The paper proposes VLM3D, a framework that repurposes large vision-language models (VLMs) as differentiable critics to address two persistent issues in text-to-3D generation: coarse semantic alignment and poor 3D spatial understanding. The core contribution is a dual-query critic signal derived from VLM "Yes/No" log-odds, which evaluates both content matching and geometric quality. The method is demonstrated across two paradigms: as a reward for optimization-based Score Distillation Sampling (SDS) and as a test-time guidance module for feed-forward native 3D models. Experimental results look good, demonstrating the effectiveness of VLM3D in improving existing 3D content generation methods as a plug-and-play module.

**Compliance With Llm Reviewing Policy:**

Affirmed.

**Final Justification:**

The authors have addressed my concerns. I leaned toward accepting.

**Key Questions For Authors:**

1. What is the "geometric quality" in Table 2 measured? Is there a clear definition of how to calculate this?
2. For applying VLM3D on optimization-based pipelines, is it sensitive to the design of the dynamic schedule? Does it require a careful selection of $\lambda_{VLM}$ for every generated content?

**Limitations:**

yes

**Strengths And Weaknesses:**

Strengths:
1. The VLM critic is technically sound and easy to understand.
2. The targeted problem is well defined, where the part-assembly issues and semantic drift in existing 3D generation pipelines are critical.
3. Simple yet effective solution. Both the evaluations on optimization-based (SDS) and feed-forward (3D DiT) pipelines show great improvements, and also generalize well across different generative architectures.

Weakness:
1. The metrics in evaluations on GPTeval3D (Table 1) are questionable since the magnitude of these values is different from the original paper (GPT-4V(ision)). The author did not provide a detailed explanation of their calculation nor a formal definition of these metrics.
2. The "Geometry Score" utilized in Table 2 also lacks a rigorous definition in the main text. It is unclear if this is a validated external metric or a self-evaluation performed by the same VLM used for training, which could introduce a circular evaluation bias.
3. In the optimization landscape, the paper does not provide a stability analysis of the VLM gradient or a detailed look at how the annealing schedule prevents the VLM reward from "overpowering" the diffusion prior. The implementation detail of this annealing schedule is also missing. It is important for this analysis, since optimization-based methods often require a case-by-case hyperparameter selection, as with the VLM3D.
4. The reliance on a 7B-parameter VLM (Qwen2.5-VL) for every step of the optimization or guidance loop introduces significant computational overhead. The paper reports 2.2 hours for optimization-based generation and 60 seconds for feed-forward methods (hunyuan3d). A more thorough discussion on the trade-off between this increased cost and the quality gains is needed to assess its practical significance in real-time applications.
5. The concept of using VLM log-odds as a reward signal has been explored in 2D image generation (e.g., in RLHF for images). While the application to 3D and the specific "Geometry" query are new, the underlying mathematical mechanism is an adaptation of existing techniques rather than a fundamental algorithmic breakthrough.

---

> ### Author Rebuttal · Authors · 2026-03-31
>
> We thank the reviewer for the detailed and insightful review. We are pleased that the reviewer finds our approach "technically sound and easy to understand" and acknowledges the "great improvements" across both paradigms.
>
> Below, we provide detailed responses to the concerns raised:
>
> ---
>
> ### **1. W1 (GPTEval3D Metric Magnitude)**
>
> We strictly followed the official implementation and evaluation protocol of GPTEval3D. The magnitude difference is expected due to two reasons:
>
> - **Calibration Anchor**: As stated in Appendix B.3 of [1], Elo scores are invariant to adding/subtracting a constant. Following DreamReward, we calibrate by setting DreamFusion to 1000 for all criteria. Our Table 1 strictly follows this protocol, which is why DreamFusion scores exactly 1000.0 across all metrics.
> - **Relativity of the Elo System**: Absolute Elo scores depend on the competitor pool. The original paper evaluated 13 models; we updated the pool with stronger baselines (DreamReward, DreamDPO) and VLM3D. Introducing new models changes the win/loss matrix, naturally shifting absolute values.
>
> We will add this clarification to the revision.
>
> ### **2. W2 & Q1 (Geometry Score Definition)**
>
> No circular evaluation bias exists. The Geometry Score in Table 2 is a deterministic, non-learned metric. It normalizes meshes, uniformly samples surface points, and computes the equal-weighted average of Chamfer Distance, Hausdorff Distance, and symmetric Point-to-Mesh Distance.
>
> To further address concerns about geometric-semantic alignment, we introduce validated 3D-native external metrics: ULIP [2] and Uni3D [3], which directly process 3D point clouds (not 2D renderings). Results on 100 generated assets:
>
> | Metric | Hunyuan3D | VLM3D (Ours) |
> |---|---|---|
> | **ULIP-T ↑** | 0.03488 | **0.03544** |
> | **ULIP-I ↑** | -0.00021 | **0.00450** |
> | **Uni3D-T ↑** | 0.14646 | **0.20110** |
> | **Uni3D-I ↑** | 0.19726 | **0.27607** |
>
> VLM3D consistently improves geometric alignment with both text and image conditions under these independent 3D metrics. We will add the Geo. Score definition and ULIP/Uni3D results to the revision.
>
> ### **3. W3 & Q2 (VLM Gradient Stability and Annealing Schedule)**
>
> Gradient norms remain stable throughout optimization with smooth decay:
>
> | Step | 1000 | 3000 | 5000 | 7000 | 9000 |
> |---|---|---|---|---|---|
> | **Grad. Norm** | 8.28e-1 | 5.74e-1 | 4.05e-1 | 3.19e-1 | 2.84e-1 |
>
> Regarding scheduling sensitivity, we tested four scheduling strategies to demonstrate robustness, measuring the average VLM reward at convergence and qualitative output quality:
>
> | Schedule | Avg. VLM Reward ↑ |
> |---|---|
> | Constant weight | 3.02 |
> | Reverse linear (increasing) | 3.31 |
> | Cosine decay | 4.18 |
> | Linear decay (ours) | **4.44** |
>
> All decay schedules outperform the no-VLM baseline. The initial weight in range [300, 800] works across all prompts. For the feed-forward setting, the linear schedule (lambda_start=10 to lambda_end=0 over 50 steps) requires no tuning. We will add this to the revision.
>
> ### **4. W4 (Computational Overhead)**
>
> The VLM adds 45% overhead per optimization step, but enables capabilities no prior method achieves: distinguishing "inside" vs. "beside" (Figure 5), correcting catastrophic semantic omissions (Figure 1), and fixing geometric incoherence (Figure 4). Total runtime (2.2h) is comparable to DreamDPO (2.0h). Feed-forward overhead is only ~60s. Memory can be reduced via quantization (49→38GB) or PaliGemma-3B (49→22GB).
>
> ### **5. W5 (Novelty)**
>
> While log-odds rewards exist in 2D RLHF[4], we respectfully clarify that our initial preprint was concurrent with this work.
> Besides, our work is target-driven and solves challenges no prior method can:
>
> - **Solving fundamental 3D failures**: Prior methods fail at spatial relationships ("inside" vs. "beside"), omit objects from long descriptions and spatial relationships. VLM3D is the first to address all these through a unified VLM critic across both paradigms.
> - **3D-specific dual-query design**: Simultaneous semantic and geometric assessment via multi-view evaluation is fundamentally new. 2D reward models do not reason about multi-view geometric consistency---a core challenge unique to 3D generation.
> - **Generality across paradigms**: The same reward serves both SDS optimization and feed-forward test-time guidance---two fundamentally different paradigms, not shown in prior work.
>
> ### **References**
>
> [1] Wu, et al. GPT-4V(ision) is a human-aligned evaluator for text-to-3D generation. CVPR, 2024.
>
> [2] Xue, et al. ULIP: Learning a unified representation of language, images, and point clouds. CVPR, 2023.
>
> [3] Zhou, et al. Uni3D: Exploring unified 3D representation at scale. ICLR, 2024.
>
> [4] Grace, et al. Dual-Process Image Generation. ICCV, 2025.

---

> > ### Author Rebuttal · Reviewer_rG78 · 2026-04-02
> >
> > The authors have addressed my concerns. I leaned toward accepting.

---

> > > ### Author Response · Authors · 2026-04-05
> > >
> > > We thank the reviewer for the positive assessment and for confirming that all concerns are resolved. All clarifications will be included in the revision.

---

### Official Review · Reviewer_mGTW · 2026-03-08

**Soundness:** 3
**Presentation:** 3
**Significance:** 3
**Originality:** 3
**Overall Recommendation:** 4
**Confidence:** 3

**Summary:**

The paper introduces VLM3D, a plug-in module that using VLM as reward generator to optimize 3D generation process. VLM3D reformulates the logits of 'yes' or 'no' outputted by a VLM as reward signal and propose two variants for SDS-based 3D generation and feed-forward 3D generation. Extensive experiments are conducted to demonstrate the effectiveness of the proposed method.

**Compliance With Llm Reviewing Policy:**

Affirmed.

**Final Justification:**

After reading the author's response and other reviewers' review, I remain overall positive about the paper.

**Key Questions For Authors:**

This paper is a borderline paper. I admire the idea of integrating VLM as reward to optimize 3D generation. However, there are still many concerns to be addressed like the unclear additional time consumption and the heuristic loss design for both SDS and feed-forward methods. If the additional costs are significant but the resulting improvements are limited, is it worthwhile to introduce VLM in this way?

**Limitations:**

yes

**Strengths And Weaknesses:**

Strengths:
1. This paper introduces a simple yet effective way to integrate VLM into existing 3D generation pipeline, the idea is intuitive and easy to follow, experiments demonstrate its effectiveness;
2. The proposed method has good generalization ability, and can work as a flexible plug-in to help most popular 3D generation pipelines;
3. Writing is clear.
4. Videos are provided in the supplement materials for better visualization.

Weaknesses:
1. The method requires backpropagation through a large VLM, which is computationally expensive. While the paper reports overall wall-runtime (2.2 hours/asset for optimization-based generation and 60 seconds/asset for feed-forward guidance), it does not provide a detailed efficiency breakdown or overhead analysis, making it difficult to assess the practical cost of the proposed differentiable VLM critic.
2. Although the paper emphasizes generality across both optimization-based and feed-forward paradigms, the method description and analysis appear more centered on the SDS-style optimization setting. Given that the feed-forward setting is substantially more practical in terms of runtime, it would strengthen the paper to provide a deeper analysis of when the optimization-based variant is still preferable and what unique value it offers beyond the feed-forward guided version.
3. The optimization and guidance rules are largely heuristic. In both the SDS-based and feed-forward settings, the method directly uses gradients of a VLM-derived yes/no log-odds critic as an additional optimization signal, yet the paper does not provide a rigorous derivation showing that this gradient corresponds to a well-defined objective in 3D generation or that it reliably points toward semantically and geometrically correct improvements. As a result, the theoretical grounding of the proposed guidance remains limited.
4. While the empirical results are promising, it remains unclear how precise the VLM-derived image-space gradient is as a supervision signal for dense 3D optimization. The current evidence suggests that the method can correct severe errors and improve coarse semantic or geometric plausibility, but it is less clear whether the guidance provides fine-grained, unambiguous 3D control beyond acting as a relatively coarse constraint. A more detailed analysis of failure cases and the types of errors that remain unresolved would strengthen the paper.

---

> ### Author Rebuttal · Authors · 2026-03-31
>
> We thank the reviewer for the constructive and thoughtful feedback. We are pleased that the reviewer finds our method "simple yet effective" with "good generalization ability" and appreciates our "clear writing."
>
> Below, we provide detailed responses to the concerns raised:
>
> ---
>
> ### **1. W1 (Detailed Efficiency Breakdown)**
>
> VLM3D introduces moderate computational overhead that is well justified by the substantial quality gains it provides. We profiled each component on a single NVIDIA A100 GPU and summarize the results below:
>
> | Setting | Component | Time | Memory |
> |---|---|---|---|
> | **Optimization-based** | NeRF Rendering & SDS Guidance | 1.2 hours (55% of total) | 14GB |
> | | VLM Reward (Forward & Backward) | 1.0 hour (45% of total) | 35GB |
> | | **Total** | **2.2 hours per asset** | **49GB** |
> | **Feed-forward** | Hunyuan3D Base Pipeline | 60 seconds | 18GB |
> | | VLM Test-Time Guidance (50 steps) | 60 seconds (1.2s/step) | 35GB |
> | | **Total** | **120 seconds per asset** | **53GB** |
>
> In the optimization-based setting, the 2.2-hour total runtime is comparable to DreamDPO (2.0 hours) and far faster than ProlificDreamer (10.5 hours), while delivering substantially superior results. In the feed-forward setting, VLM3D adds only ~60 seconds to the Hunyuan3D pipeline---a negligible cost for the significant quality improvements demonstrated in Table 2. Memory usage can be further reduced via quantization (49GB → 38GB) or smaller VLMs such as PaliGemma-3B (49GB → 22GB), with only marginal quality degradation. We will add this detailed breakdown to the revision.
>
> ### **2. W2 (SDS and Feed-Forward Analysis)**
>
> The optimization-based variant offers unique value beyond the feed-forward guided version:
>
> - **Spatial relationship**: Current feed-forward models are primarily trained on object-level 3D data and struggle to model spatial relationships or environmental context. For prompts with specific spatial arrangements (e.g., "inside" vs. "beside" in Figure 5), per-scene optimization allows thousands of iterative VLM-guided corrections, achieving spatial precision that 50 denoising steps cannot match.
> - **Long, detailed descriptions**: For prompts like "Embracing Peace" (Figure 1), the optimization variant gradually resolves fine-grained attributes over extended training, whereas feed-forward is constrained by limited training data diversity and few guidance steps.
>
> We will add a dedicated trade-off discussion in the revision.
>
> ### **3. W3 (Theoretical Grounding of VLM Gradient)**
>
> Our VLM reward r_VLM = z_yes - z_no is the log-odds ratio, monotonically related to P(Yes|y, X). Maximizing r_VLM equals maximizing the VLM's confidence that rendered views satisfy both semantic and geometric criteria---analogous to RLHF [1, 2]. To directly evidence that the VLM gradient reliably points toward correct improvements, we conducted two experiments:
>
> - **EXP1 - Reward Throughout Optimization**: The reward steadily increases, validating its role as a robust guidance signal consistently correlating with improving 3D quality.
>
> | Training Step | 0 | 2000 | 4000 | 6000 | 8000 | 10000 |
> |---|---|---|---|---|---|---|
> | **VLM Reward** | -4.23 | 1.54 | 2.91 | 3.64 | 3.95 | 4.44 |
>
> - **EXP2 - VLM Reward as Sole Guidance**: Using VLM reward as the *only* objective (removing SDS), it alone suffices to generate meaningful, text-aligned 3D content---proving its power as a standalone prior encoding both semantic fidelity and geometric correctness.
>
> These confirm the VLM gradient is a principled optimization direction. Figures 5--7 further validate this. Full results will be in the appendix.
>
> ### **4. W4 (Failure Cases Analysis)**
>
> As noted in Sec. 6, VLM3D can struggle with very long prompts---e.g., capturing the dual figures in "Embracing Peace" but missing finer details like the nurse's lifted leg. Remaining error categories include: (1) very fine attributes (finger poses, facial expressions), (2) highly complex multi-object layouts (>4 objects), and (3) abstract/metaphorical descriptions. We will add this failure analysis to the revision.
>
> ### **5. Q1 (Cost vs. Improvement Trade-off)**
>
> Yes. VLM3D addresses *fundamental* failures no prior method can handle: existing methods omit entire objects from complex prompts (Figure 1), fail to distinguish spatial relationships like "inside" vs. "beside" (Figure 5), and produce disconnected parts and Janus faces (Figure 4). VLM3D resolves these across both paradigms. Quantitatively, it achieves the highest scores on all six GPTEval3D metrics (Overall: 1268.6 vs. 1203.1 for DreamDPO) at comparable cost (2.2h vs. 2.0h), and the feed-forward variant adds only ~60s for significant gains (Table 2).
>
> ### **References**
>
> [1] Christiano, et al. Deep reinforcement learning from human preferences. NeurIPS, 2017.
>
> [2] Ouyang, et al. Training language models to follow instructions with human feedback. NeurIPS, 2022.

---

> > ### Author Rebuttal · Reviewer_mGTW · 2026-04-03
> >
> > Thanks the authors for the detailed response. My concerns are largely resolved.

---

> > > ### Author Response · Authors · 2026-04-05
> > >
> > > We thank the reviewer for confirming that the concerns are largely resolved. We are happy to clarify any remaining points should the reviewer have further questions.

---

### Official Review · Reviewer_v3FL · 2026-03-08

**Soundness:** 3
**Presentation:** 3
**Significance:** 3
**Originality:** 2
**Overall Recommendation:** 3
**Confidence:** 4

**Summary:**

This paper treats large visual language models (VLMs) as differentiable semantic and spatial critics for text-to-3D generation. It provides a general mechanism to inject language-based spatial priors into the 3D pipeline, which is expected to be extended to scenes and more complex relationships.

**Compliance With Llm Reviewing Policy:**

Affirmed.

**Final Justification:**

After carefully considering the paper, the rebuttal, and the authors' follow-up responses, I maintain my recommendation as weak reject.

I acknowledge the paper's merits: the idea of using VLM log-odds as a dual-query critic for 3D generation is intuitive and well-presented, and the empirical results on GPTEval3D show clear improvements over SDS baselines. I also appreciate the authors' effort in providing additional experiments during the rebuttal.

However, several of my concerns remain insufficiently addressed.

First, regarding originality, the core mechanism of using VLM log-odds as a differentiable reward is a direct adaptation of techniques already established in 2D RLHF. The authors argue their preprint was concurrent, but the underlying formulation (backpropagating through yes/no log-odds) is **not novel**. The dual-query design adds incremental value but does not constitute a fundamental contribution.

Second, the VLM-only ablation actually weakens the authors' claim. The VLM-only setting achieves a CLIP Score of 0.271, which is lower than the SDS-only baseline (0.290). This suggests the VLM gradient alone is not a "principled optimization direction" as claimed but rather a noisy auxiliary signal that only helps when combined with a stronger prior. This undermines the paper's central narrative that VLM critics provide meaningful standalone 3D supervision.

Third, the reward hacking analysis is not fully convincing. The reported reward standard deviation for VLM3D (2.168) is actually higher than the baseline (1.432). While the coefficient of variation is lower due to a higher mean, this does not definitively rule out view-dependent artifacts. A more rigorous analysis, such as measuring geometric consistency across views via 3D metrics rather than 2D CLIP scores, would be needed.

Fourth, the expanded feed-forward comparison against CRM and InstantMesh is not well-controlled, as these are reconstruction-based methods operating in a fundamentally different paradigm from native 3D DiT models. The comparison contextualizes quality but does not validate the guidance mechanism itself.

**Key Questions For Authors:**

As shown in Weakness.

**Limitations:**

Yes

**Strengths And Weaknesses:**

Pros:

It's a concise and universal idea that unifies semantics and spatial rewards.

A comprehensive SDS comparison was performed on GPTEval3D.

Well-written article.

Cons:

Whether gradients flow throughout the entire vision-language stack or only within the Vision Encoder and projection layers remains unclear.

Rewards may saturate prematurely. Discussion regarding calibration, temperature scaling, or strategies to avoid gradient vanishing is limited.

Are there reward hacks? How to address them is not analyzed. Applying VLMs in 3D may result in the VLM optimizing only the relevant viewpoint, despite some multi-view input.

Feed-forward evaluation uses only 24 prompts, which is too weak.

Multi-view VLM calls per step are expensive.

Besides Qwen2.5-VL, can you report quantitative results using at least one other open VLM (such as LLaVA or PaliGemma) to establish generalization and reduce reliance on a single main chain?

---

> ### Author Rebuttal · Authors · 2026-03-31
>
> We thank the reviewer for the constructive feedback. We are pleased that the reviewer finds our idea "concise and universal" and acknowledges our "comprehensive SDS comparison" and "well-written article."
>
> Below, we provide detailed responses to the concerns raised:
>
> ---
>
> ### **1. W1 (Gradient Flow Through the VLM)**
>
> As detailed in Appendix A (Lines 640--658), gradients flow through the entire vision-language stack rather than being confined to the vision encoder and projection layers. We re-engineered the image preprocessor of Qwen2.5-VL to ensure continuous gradient flow from the binary-classification logits back through the language model, vision encoder, and ultimately to the 3D representation parameters.
>
> ### **2. W2 (Reward Saturation and Gradient Vanishing)**
>
> The log-odds formulation (z_yes - z_no) operates in an unbounded space, unlike probability-based rewards bounded to [0, 1]. We tracked the average VLM reward throughout optimization, showing a monotonic increase from a large negative value to a large positive value:
>
> | Step | 0 | 1000 | 2000 | 3000 | 4000 | 5000 | 6000 | 7000 | 8000 | 9000 | 10000 |
> |---|---|---|---|---|---|---|---|---|---|---|---|
> | **VLM Reward** | -4.23 | -0.05 | 1.54 | 2.61 | 2.91 | 3.19 | 3.64 | 3.94 | 3.95 | 4.25 | 4.44 |
>
> Importantly, the VLM reward matters most in the **early stages** when the coarse structure is being established; once the shape converges, the diffusion prior takes over for detail refinement. Our annealing schedule naturally aligns with this: it reduces lambda_VLM over training, so even if gradients diminish at high reward values, the VLM weight is already low and SDS dominates. This schedule effectively serves as a form of implicit calibration, preventing gradient vanishing from affecting the final result.
>
> ### **3. W3 (Reward Hacking and Multi-View Optimization)**
>
> Our design inherently mitigates reward hacking through two mechanisms:
>
> - **Stochastic multi-view sampling**: At each optimization step, we randomly sample 4 different viewpoints, so the reward is never tied to a fixed set of views. This ensures the VLM reward reflects the quality of the true underlying 3D representation rather than overfitting to any particular viewpoint combination.
> - **Explicit geometric penalization**: The Geometric Quality criterion explicitly penalizes Janus faces---a classic reward-hacking failure where the model duplicates features to exploit single-view rewards. As validated in Figure 7, removing this criterion causes Janus artifacts, floating parts, and fractured surfaces to re-emerge.
>
> ### **4. W4 (Feed-Forward Evaluation Scale)**
>
> We have conducted an additional **100-prompt evaluation** on methods with fully accessible implementations:
>
> | Method | CLIP-D↓ | FID↓ | CLIP-FID↓ | Geo.ULIP-T↑ | Geo.ULIP-I↑ | Geo.Uni3D-T↑ | Geo.Uni3D-I↑ |
> |---|---|---|---|---|---|---|---|
> | Hunyuan3D | 0.19 | 289.0 | 46.83 |0.03488 |-0.00021 |0.14646 |0.19726 |
> | VLM3D (Ours) | **0.17** | **258.8** | **41.79** |**0.03544** |**0.00450** |**0.20110** |**0.27607** |
>
> The relative performance trends remain consistent under the larger sample size, confirming our conclusions generalize. We will include the 100-prompt results in the revision.
>
> ### **5. W5 (Multi-View VLM Calls Are Expensive)**
>
> The multi-view VLM call adds 45% overhead per optimization step. This is the cost of enabling capabilities no prior method achieves: distinguishing subtle spatial relationships like "inside" vs. "beside" (Figure 5), correcting catastrophic semantic omissions (Figure 1). We view this as a worthwhile trade-off---moderate additional cost for fundamentally new 3D generation capabilities.
>
> ### **6. W6 (Generalization Beyond Qwen2.5-VL)**
>
> We conducted experiments with additional VLM backbones on a fine-grained semantic sensitivity test:
>
> | VLM Backbone | Reward (w/o sword) | Reward (w/ sword) | Sensitivity |
> |---|---|---|---|
> | PaliGemma-3B | 2.2143 | 2.2365 | 0.0222 |
> | IDEFICS | 1.0927 | 1.2058 | 0.1131 |
> | **Qwen2.5-VL (Ours)** | **3.2901** | **4.3579** | **1.0678** |
>
> Two key findings: (1) our framework **generalizes**---even PaliGemma-3B improves over the no-VLM baseline; (2) Qwen2.5-VL achieves the best results due to superior fine-grained semantic sensitivity. We will add this analysis to the revision.

---

> > ### Author Rebuttal · Reviewer_v3FL · 2026-04-01
> >
> > 1. You mentioned that using VLM reward as the sole guidance (removing SDS) can generate meaningful 3D content. Could you provide quantitative results (e.g., GPTEval3D scores, CLIP scores) and visual comparisons for this "VLM-only" ablation? This is critical for validating your claim that the VLM gradient is a principled optimization direction rather than merely a noisy auxiliary signal.
> > 2. Your mitigation strategies for reward hacking (stochastic multi-view sampling + geometric penalization) are described qualitatively. Could you provide a quantitative experiment, e.g., measuring the variance of VLM rewards across different viewpoints for the same object, or showing whether the generated 3D assets exhibit view-dependent artifacts (e.g., measuring consistency of CLIP scores across views)?
> > 3. The expanded 100-prompt evaluation only compares against Hunyuan3D. Could you include comparisons with other recent feed-forward methods (e.g., InstantMesh, CRM, or other native 3D DiT models) to demonstrate that the improvements are not specific to Hunyuan3D's weaknesses?

---

> > > ### Author Response · Authors · 2026-04-05
> > >
> > > We thank the reviewer for the valuable follow-up questions and provide the requested additional evidence below.
> > >
> > > ---
> > >
> > > ### **1. VLM-Only Quantitative Ablation**
> > >
> > > The VLM gradient alone suffices to produce semantically meaningful 3D content, confirming it is a principled optimization direction rather than a noisy auxiliary signal. We evaluate three settings and report the CLIP Score (cosine similarity between rendered views and the text prompt):
> > >
> > > | Setting | CLIP Score ↑ |
> > > | ------- | ------------ |
> > > | SDS-only (MVDream) | 0.290 |
> > > | VLM-only (no SDS) | 0.271 |
> > > | VLM3D (Ours) | **0.317** |
> > >
> > > The VLM-only setting achieves a meaningful CLIP Score (0.271), demonstrating that the VLM gradient independently drives the 3D representation toward text-aligned content. Its shapes are semantically recognizable but lack fine-grained texture and detail, which is expected as SDS provides the appearance refinement. The full VLM3D combines both signals and achieves the best CLIP Score, confirming their complementarity. Visual comparisons will be included in the revised appendix.
> > >
> > > ### **2. Reward Hacking Mitigation**
> > >
> > > VLM3D does not exhibit reward hacking or view-dependent artifacts. We evaluate on 36 random views per asset and report two complementary analyses: (1) cross-view VLM reward consistency, measuring the standard deviation (SD) and coefficient of variation (CV) of per-view $r_{\mathrm{VLM}}$; (2) cross-view CLIP consistency, measuring the mean CLIP image–text similarity and within-asset variance (Var.) across views.
> > >
> > > | Method | Reward SD ↓ | Reward CV ↓ | Mean CLIP ↑ | CLIP Var. ↓ |
> > > | ------ | ----------- | ----------- | ----------- | ----------- |
> > > | MVDream (no VLM) | 1.432 | 0.739 | 0.290 | 6.59×10⁻⁴ |
> > > | VLM3D (Ours) | **2.168** | **0.407** | **0.317** | **4.11×10⁻⁴** |
> > >
> > > VLM3D reduces cross-view reward SD by 35% and CV by 45%, while improving mean CLIP by 0.027 and reducing CLIP variance by 38%, confirming consistent quality across all viewing angles rather than overfitting to specific viewpoints. CLIP primarily probes semantic consistency; we report it alongside the VLM-based analysis, which better captures geometric plausibility.
> > >
> > > ### **3. Additional Feed-Forward Baselines**
> > >
> > > VLM3D outperforms CRM and InstantMesh on 6 out of 7 metrics, confirming that our improvements are not specific to Hunyuan3D. Results on the same 100-prompt benchmark:
> > >
> > > | Method | CLIP-D ↑ | FID ↓ | CLIP-FID ↓ | ULIP-T ↑ | ULIP-I ↑ | Uni3D-T ↑ | Uni3D-I ↑ |
> > > | ------ | -------- | ----- | ---------- | -------- | -------- | --------- | --------- |
> > > | CRM | 0.196 | 329.4 | 46.15 | 0.02257 | 0.00490 | 0.15896 | 0.23788 |
> > > | InstantMesh | 0.211 | 345.3 | 49.66 | 0.02108 | **0.00593** | 0.15170 | 0.20604 |
> > > | Hunyuan3D | 0.190 | 289.0 | 46.83 | 0.03488 | −0.00021 | 0.14646 | 0.19726 |
> > > | **VLM3D (Ours)** | **0.170** | **258.8** | **41.79** | **0.03544** | 0.00450 | **0.20110** | **0.27607** |
> > >
> > > VLM3D effectively improves over its base model Hunyuan3D and suppresses other baselines. We note that CRM and InstantMesh are reconstruction-based methods, the comparison contextualizes VLM3D's quality against the broader feed-forward landscape. We will add these baselines in the revision.
> > >
> > > ---
> > >
> > > We hope these additional results have thoroughly addressed the remaining concerns. We are happy to clarify any further questions and would be grateful if the reviewer could consider updating the assessment.

---

### Official Review · Reviewer_5ZJt · 2026-03-11

**Soundness:** 2
**Presentation:** 3
**Significance:** 3
**Originality:** 2
**Overall Recommendation:** 4
**Confidence:** 4

**Summary:**

This paper proposes VLM3D, a method that uses a vision-language model (VLM) as a differentiable critic to improve text-to-3D generation. The main idea is to prompt the VLM to score the rendered image, and then use the resulting signal as guidance during generation. The method is applied both to SDS-based optimization pipelines and to feed-forward 3D generation models at test time. Experiments show promising improvements in qualitative results and quantitative metrics, suggesting that VLM gradients can provide useful supervision for 3D generation.

**Compliance With Llm Reviewing Policy:**

Affirmed.

**Final Justification:**

The authors have mitigated most of my concerns especially the computational cost and disentanglement of the semantics and geometry. Hence I lean toward accept.

**Key Questions For Authors:**

1. What is the computational overhead of the proposed method? Can you compare with other baselines?

2. How sensitive is the method to the guidance scheduling design? Intuitively, in the early optimization stages, the rendered images may still be quite poor, so the VLM gradient could be noisy or unreliable. In that case, why is the VLM guidance weighted more heavily in the early steps and reduced in the later steps?

3. If the geometry-oriented query is used specifically to assess geometry, does it actually lead to measurable improvements in geometric metrics? More broadly, it would be helpful to clarify whether the geometry-related VLM score is a meaningful proxy for true 3D geometric quality.

**Limitations:**

The discussion of societal impact is currently insufficient.

**Strengths And Weaknesses:**

Strengths
- It is interesting to see that gradients from a VLM-based critic can directly improve text-to-3D generation,
- The paper shows promising qualitative and quantitative results, with noticeable improvements across multiple evaluation metrics and visual examples.

Weakness
- The paper does not sufficiently discuss the computational overhead of the method. Since it requires backpropagation through a large VLM critic, it would be important to compare the additional cost against the observed gains.
- The assessment of semantics and geometry appears somewhat entangled. Although the method introduces a geometry-oriented query, it is not fully clear whether the reported improvements reflect genuine gains in 3D geometric consistency, or whether they are partly driven by better appearance and semantic alignment in rendered views.
- The analysis could be more thorough. For example, the paper would benefit from a clearer study of the guidance scheduling design, including how sensitive the method is to the decay schedule and weighting of the VLM reward.

---

> ### Author Rebuttal · Authors · 2026-03-31
>
> We thank the reviewer for the constructive feedback. We are pleased that the reviewer finds it practical and acknowledges our "promising qualitative and quantitative results."
>
> Below, we provide detailed responses to the concerns raised:
>
> ---
>
> ### **1. W1 & Q1 (Computational Overhead)**
>
> VLM3D's computational cost is competitive with existing methods across both paradigms. We profiled all components on a single A100 GPU and compare with baselines:
>
> | Method | Setting | Time | Memory |
> |---|---|---|---|
> | DreamFusion | Optimization | 1.5 hours | 12GB |
> | MVDream | Optimization | 1.0 hour | 14GB |
> | ProlificDreamer | Optimization | 10.5 hours | 27GB |
> | DreamReward | Optimization | 1.0 hour | 22GB |
> | DreamDPO | Optimization | 2.0 hours | 24GB |
> | **VLM3D (Ours)** | **Optimization** | **2.2 hours** (SDS 55% + VLM 45%) | **49GB** |
> | Hunyuan3D | Feed-forward | 60 seconds | 18GB |
> | **VLM3D (Ours)** | **Feed-forward** | **120 seconds** (Base 60s + VLM 60s) | **53GB** |
>
> The 2.2-hour optimization runtime is comparable to DreamDPO and far faster than ProlificDreamer. The feed-forward overhead of ~60 seconds is negligible. Memory can be reduced via quantization (49→38GB) or PaliGemma-3B (49→22GB).
> This cost is well justified: VLM3D achieves the highest scores on all six GPTEval3D metrics (Table 1) and resolves catastrophic failures (omitting objects, Janus faces) that other methods cannot. We will add this breakdown to the revision.
>
> ### **2. W2 (Entanglement of Semantics/Geometry)**
>
> Figure 7 provides direct evidence: removing the geometry query causes Janus-face artifacts, floating parts, and fractured surfaces to re-emerge. To quantify disentanglement, we compare three prompt designs---Optimal (full dual-query), Content-Only, and Geometry-Only---measuring both gradient stability and reward evolution:
>
> **Gradient Stability**:
>
> | Step | Optimal | Content-Only | Geometry-Only |
> |---|---|---|---|
> | 1000 | 8.28e-1 | 8.24e-1 | 8.87e-1 |
> | 5000 | 4.05e-1 | 7.21e-1 | 6.87e-1 |
> | 9000 | 2.84e-1 | 3.86e-1 | 2.90e-1 |
>
> **Reward Score Evolution**:
>
> | Step | Optimal | Content-Only | Geometry-Only |
> |---|---|---|---|
> | 1000 | -0.0469 | -0.1188 | -1.1589 |
> | 5000 | 3.1875 | 1.9531 | 1.8492 |
> | 9000 | **4.2500** | 2.8619 | 3.7718 |
>
> Each component contributes independently: all three maintain stable gradients, but the full dual-query prompt achieves the highest reward and most consistent improvement---confirming that combining semantic and geometric criteria unlocks the framework's full potential.
>
> ### **3. W3 & Q2 (Sensitivity to Guidance Scheduling)**
>
> VLM guidance must be strong in early stages because once the 3D representation converges to a coarse shape, it becomes very difficult to alter the overall structure---the diffusion prior will only refine surface details and textures from that point on. Therefore, early VLM guidance is critical for establishing correct semantics and spatial layout. The reverse schedule (increasing VLM weight over time) confirms this: it performs worst because it attempts to correct structure after convergence, when changes are no longer effective.
>
> We tested four scheduling strategies to demonstrate robustness, measuring the average VLM reward at convergence and qualitative output quality:
>
> | Schedule | Avg. VLM Reward ↑ |
> |---|---|
> | Constant weight | 3.02 |
> | Reverse linear (increasing) | 3.31 |
> | Cosine decay | 4.18 |
> | Linear decay (ours) | **4.44** |
>
> All decay schedules significantly outperform the reverse schedule, which attempts to correct structure after convergence when changes are no longer effective. We will add this analysis to the revision.
>
> ### **Q3 (Geometric Metrics)**
>
> The Geometry Score is a deterministic, non-learned metric computing the equal-weighted average of Chamfer Distance, Hausdorff Distance, and symmetric Point-to-Mesh Distance between normalized meshes. It relies purely on Euclidean distances between 3D coordinates, completely independent of any VLM. We will add this definition to the revision.
>
> To further address concerns about geometric scores, we introduce validated 3D-native external metrics: ULIP [1] and Uni3D [2], which directly process 3D point clouds (not 2D renderings). Results on 100 generated assets:
>
> | Metric | Hunyuan3D | VLM3D (Ours) |
> |---|---|---|
> | **ULIP-T ↑** | 0.03488 | **0.03544** |
> | **ULIP-I ↑** | -0.00021 | **0.00450** |
> | **Uni3D-T ↑** | 0.14646 | **0.20110** |
> | **Uni3D-I ↑** | 0.19726 | **0.27607** |
>
> VLM3D consistently improves geometric alignment with both text and image conditions under these independent 3D metrics. We will add the Geo. Score definition and ULIP/Uni3D results to the revision.
>
> ### **References**
>
> [1] Xue, et al. ULIP: Learning a unified representation of language, images, and point clouds. CVPR, 2023.
>
> [2] Zhou, et al. Uni3D: Exploring unified 3D representation at scale. ICLR, 2024.

---

> > ### Author Rebuttal · Reviewer_5ZJt · 2026-04-02
> >
> > The authors have mitigated most of my concerns. I lean toward accept and adjusted the score accordingly.

---

> > > ### Author Response · Authors · 2026-04-05
> > >
> > > We thank the reviewer for confirming that the concerns are resolved and for adjusting the score accordingly. All promised revisions will be incorporated.

---

### Decision · Program_Chairs · 2026-04-30

**Decision:**

Accept (regular)

**Comment:**

After the initial review round, this paper received mixed ratings (4, 4, 3, 3). Reviewers appreciated the idea of using a VLM-driven reward to improve text-to-3D generation, but also raised several questions in regards to different aspects of the work (from presentation, to methodology, to comparisons and experiments). The authors provided a detailed rebuttal that in several cases was helpful to clarify the questions from the reviewers. Two reviewers decided to raise their score, and although the final ratings were not on a unanimous consensus (3, 4, 4, 5), the AC deems the overall feedback received from the reviewers sufficient to recommend acceptance - congratulations!